



# Observation-Based Constraints on Modeled Aerosol Surface Area: Implications for Heterogeneous Chemistry

Rachel A. Bergin[1], Monica Harkey[2,3], Alicia Hoffmann[2,3], Richard H. Moore[4], Bruce Anderson[4], Andreas Beyersdorf[4,*], Luke Ziemba[4], Lee Thornhill[4,5], Edward Winstead[4,5], Tracey Holloway[2,3], and Timothy H. Bertram[1]

[1]Department of Chemistry, University of Wisconsin - Madison, Madison, WI 53703, USA
[2]Nelson Institute Center for Sustainability and the Global Environment, University of Wisconsin–Madison, Madison, WI 53703, USA
[3]Department of Atmospheric and Oceanic Sciences, University of Wisconsin–Madison, Madison, WI 53703, USA
[4]NASA Langley Research Center, Hampton, VA 23666, USA
[5]Science Systems and Applications, Inc., Hampton, VA 23666, USA
[*]Now at California State University, San Bernardino, San Bernardino, CA 92407

*Correspondence to*: T.H. Bertram (timothy.bertram@wisc.edu)

**Abstract.** Heterogeneous reactions occurring at the surface of atmospheric aerosol particles regulate the production and lifetime of a wide array of atmospheric gases. Aerosol surface area plays a critical role in setting the rate of heterogeneous reactions in the atmosphere. Despite the central role for aerosol surface area, there are few assessments of the accuracy of aerosol surface area concentrations in regional and global models. In this study, we compare aerosol surface area concentrations in the EPA's Community Multiscale Air Quality (CMAQ) model with commensurate observations from the 2011 NASA flight-based DISCOVER-AQ (Deriving Information on Surface Conditions from COlumn and VERtically Resolved Observations Relevant to Air Quality) campaign. The study region includes the Baltimore and Washington, DC metropolitan area. Dry aerosol surface area was measured aboard the NASA P-3B aircraft using an Ultra-High Sensitivity Aerosol Spectrometer (UHSAS). We show that modeled and measured dry aerosol surface area, $S_{a,\mathrm{mod}}$ and $S_{a,\mathrm{meas}}$ respectively, are modestly correlated ($r^2 = 0.52$) and on average agree to within a factor of two ($S_{a,\mathrm{mod}}/S_{a,\mathrm{meas}} = 0.44$) over the course of the 13 research flights. We show that $S_{a,\mathrm{mod}}/S_{a,\mathrm{meas}}$ does not depend strongly on photochemical age or the concentration of secondary biogenic aerosol, suggesting that the condensation of low-volatility gas-phase compounds does not strongly affect model-measurement agreement. In comparison, there is strong agreement between measured and modeled aerosol number concentration ($N_{\mathrm{mod}}/N_{\mathrm{meas}} = 0.87$, $r^2 = 0.63$). The persistent underestimate of $S_a$ in the model, combined with strong agreement in modeled and measured aerosol number concentrations, suggests that model representation of the size distribution of primary emissions or secondary aerosol formed at the early stages of oxidation may contribute to the observed differences.

For reactions occurring on small particles, the rate of heterogeneous reactions is a linear function of both $S_a$ and the reactive uptake coefficient ($\gamma$). To assess the importance of uncertainty in modeled $S_a$ for the representation of heterogeneous reactions in models, we compare both the mean and the variance in $S_{a,\mathrm{mod}}/S_{a,\mathrm{meas}}$ to that in $\gamma(N_2O_5)_{\mathrm{mod}}/\gamma(N_2O_5)_{\mathrm{meas}}$. We find that the





uncertainty in model representation of heterogeneous reactions is primarily driven by uncertainty in the parametrization of reactive uptake coefficients, although the discrepancy between $S_{a,\text{mod}}$ and $S_{a,\text{meas}}$ is not insignificant. Our analysis suggests that model improvements to aerosol surface area concentrations, in addition to more accurate parameterizations of heterogeneous kinetics, will advance the representation of heterogeneous chemistry in regional models.

## 1 Introduction

### 1.1 The Role of Aerosol Surface Area in Heterogeneous Reaction Kinetics

Reactions occurring at atmospheric interfaces, such as suspended aerosol particles, catalyze the production and loss of key gas-phase compounds in Earth's atmosphere with important implications for regional air quality (Chang et al., 2011). The rate of heterogeneous reactions occurring at the surface of aerosol particles is a function of the gas-aerosol collision frequency and the per collision reaction probability. Variability in gas-aerosol collision frequency is determined by the aerosol surface area

concentration. The probability of reaction, or the net reactive uptake coefficient ($\gamma$), is reaction specific and dependent on chemical kinetics, gas accommodation at the surface, and near surface diffusion (Abbatt et al., 2012). Collectively, the first-order removal rate of a gas-phase species ($A$) from the atmosphere can be written as:

$$\frac{d[A]}{dt} = -k_{het}[A] \qquad\qquad \text{E1}$$

where the heterogeneous reaction rate constant ($k_{het}$), in the absence of gas-phase diffusion limitations, can be written as:

$$k_{het} = \frac{\gamma_{species}\omega_{species}S_a}{4} \qquad\qquad \text{E2}$$

where $\omega$ is the mean molecular speed of the gas-phase molecule (m s$^{-1}$) and $S_a$ is the surface area concentration of aerosol particles (m$^2$ m$^{-3}$).

To date, most evaluations of the role of heterogeneous chemistry on gas-phase composition have focused on uncertainty in

parameterizations of reactive uptake coefficients, such as the reactive uptake of dinitrogen pentoxide ($N_2O_5$) due to its role as a NO$_x$ sink (Brown et al., 2009; Evans and Jacob, 2005; MacIntyre and Evans, 2010; McDuffie et al., 2018). In comparison, there has been less focus on model representation of aerosol surface area concentrations, despite the fact that $k_{het}$ is linearly dependent on $S_a$. An accurate representation of aerosol surface area in regional and global chemical transport models is challenging, as $S_a$ is a complex function of size-dependent aerosol particle emissions, chemical transformations, and removal



processes. Here, we directly compare aerosol surface area concentrations in a regional chemical transport model with

commensurate aircraft measurements to assess the representation of $S_a$ in regional air quality models.

## 1.2 Calculation of Aerosol Surface Area in Regional Air Quality Models

The total aerosol particle surface area concentration has been calculated in air quality models using a variety of approaches,

including the discrete representation of the particle size distribution in defined size ranges, known as the sectional method

(Adams and Seinfeld, 2002; Gelbard et al., 1980; Jacobson, 2001; Lee et al., 2009; Lee and Adams, 2012; Luo and Yu, 2011;

Spracklen et al., 2006; Trivitayanurak et al., 2008; Yu and Luo, 2009), and a continuous modal representation of the particle

size distribution (Kleeman et al., 1997; Mann et al., 2010; Meng, 1998; Pringle et al., 2010; Sartelet et al., 2006; Stier et al.,

2005; Vignati et al., 2004; Zhang et al., 2010a). Here, we review the modal representation of particle size distributions

implemented in the Community Multiscale Air Quality (CMAQ) model and the calculation of both wet and dry total aerosol

surface area ($S_a$). Aerosol particle size distributions in CMAQ follow the method developed for the Regional Particulate Model,

an extension of the Regional Acid Deposition Model (Binkowski, 1999; Binkowski and Roselle, 2003) where the total particle

size distribution is treated as the superposition of three separate lognormal distributions (or modes); Aitken, accumulation, and

coarse modes (Binkowski, 1999; Whitby, 1978). The lognormal particle size distribution for each mode is defined as:


$$n(\ln D) = \frac{N}{\sqrt{2\pi}\ln\sigma_g}\exp\left[-0.5\left(\frac{\ln\frac{D}{D_g}}{\ln\sigma_g}\right)^2\right] \qquad\qquad \text{E3}$$

where $N$ is the total number concentration, $D$ is the particle diameter, and $D_g$ and $\sigma_g$ are the geometric mean diameter and

geometric standard deviation. Under this definition, the Aitken mode describes aerosol particles of diameter smaller than

approximately 0.1μm with a median diameter of 0.03μm, while the accumulation mode encompass the diameter range of 0.1

to 2.5μm with a median diameter of 0.3μm (Binkowski, 1999). The coarse mode describes particles of diameter 0.3 to about

10μm with a median diameter of 6μm. It should be noted that there is uncertainty in the exact size distributions within CMAQ

dependent on emissions parameters, so the median diameters within modes are approximate (Elleman and Covert, 2010).





Three integral properties of the aerosol size distribution are calculated in CMAQ: the zeroth ($M_0$), second ($M_2$), and third

moments ($M_3$), where the $k^{th}$ moment of the size distribution is calculated as:

$$M_k = \int_{-\infty}^{\infty} D^k (lnD) d(lnD) = ND_g^k exp\left[\frac{k^2}{2} ln^2 \sigma_g\right] \hspace{2cm} \text{E4}$$

In this representation, $N = M_0$, $S_a = \pi M_2$, and $V = \left(\frac{\pi}{6}\right)M_3$, where $V$ is the total aerosol volume (Binkowski, 1999; Binkowski

and Roselle, 2003). Though $M_2$ is utilized in CMAQ's aerosol subroutines, it is multiplied by $\pi$ prior to use in main CMAQ

routines, such that it is identified as a modal surface area (Binkowski and Roselle, 2003).


The time rate of change of each moment is calculated for each grid box and time interval as:

$$\frac{\partial M_k}{\partial t} = P_k - L_k M_k \hspace{2cm} \text{E5}$$

where $P$ and $L$ represent the production and loss of $M_k$ in each aerosol mode. With respect to $S_a$ ($S_a = \pi M_2$), neglecting

transport terms, $P_2$ includes new particle formation (Aitken mode only), condensational growth, and primary emissions and $L_2$

includes intramodal coagulation, dry and wet deposition.

In the interpretation of model $S_a$, the following model specific details should be considered: 1) Fine particles (Aitken and

accumulation modes) do not coagulate with coarse mode particles and coarse mode particles do not coagulate with each other

(Binkowski and Roselle, 2003). 2) The size distribution for primary PM$_{2.5}$ emissions are assumed to have a geometric mean

($D_g = 0.3\mu m$) and geometric standard deviation ($\sigma_g = 2$) and > 99% of PM$_{2.5}$ emission are assigned to the accumulation mode

(Binkowski and Roselle, 2003), which may have consequent effects on the aerosol surface area distribution. 3) Particles are

assumed to be spherical.

The condensation of water is accounted for in the chemical evolution of $M_2$, thus $M_2$ is inherently the wet second moment

($M_2^w$) which is used in the calculation of heterogeneous chemical reactions. In addition to $M_2^w$, a dry second moment ($M_2^d$) is

calculated as a function of the third moment ($M_3$) as:


$$M_2^d = M_2^w \left( \frac{M_3^d}{M_3^w} \right)^{\frac{2}{3}}$$
E6

In the following analyses, we concentrate on the comparison of modeled and measured $S_a$ to evaluate the relative uncertainty
associated with model descriptions of heterogeneous kinetic mechanisms (i.e. reactive uptake coefficients, $\gamma$) and aerosol
particle size distributions (i.e. aerosol surface area, $S_a$) that combined dictate the fate of reactive gas-phase molecules.

### 1.3 Previous Model-Measurement Comparisons of Aerosol Surface Area

Evaluation of regional air quality models has largely focused on criteria air pollutants such as ozone ($O_3$) and particle mass
(e.g. $PM_{2.5}$) (Appel et al., 2021). Previous model evaluation of particle mass has focused on an array of metrics including mass
concentration (Gantt et al., 2012; Spak and Holloway, 2009; Wang et al., 2009), number concentration (Park et al., 2006;
Ranjithkumar et al., 2021; Wang et al., 2009; Zhang et al., 2010b), size distribution (Kelly et al., 2011; Nolte et al., 2015; Park
et al., 2006; Zhang et al., 2010b), composition (Knote et al., 2011; Nolte et al., 2015; Prank et al., 2016), and aerosol optical
depth (Ghan et al., 2001; Knote et al., 2011), among others. There has been a very long and detailed history of CMAQ
evaluation of $PM_{2.5}$, including both ground-based (Baker et al., 2018; Fan et al., 2005; Ghim et al., 2017; Hogrefe et al., 2009,
2015; Liu and Zhang, 2011; Prank et al., 2016; Smyth et al., 2006; Wang et al., 2021; Yu et al., 2012, 2008b, 2008a; Zhang et
al., 2019, 2006, 2010c), ship-based (Yu et al., 2012), and aircraft-based measurements (Baker et al., 2018; Chen et al., 2020;
Yu et al., 2012). For fifteen studies comparing ground-based measurements of $PM_{2.5}$ to CMAQ outputs between 1999 and
2018, ten saw an underestimation of $PM_{2.5}$ by the model ranging between 6-75% (Ghim et al., 2017; Liu and Zhang, 2011;
Prank et al., 2016; Wang et al., 2021; Yu et al., 2008a, 2012, 2008b; Zhang et al., 2019, 2006, 2010c), dependent on pollution
events and rural versus urban location, while four found that CMAQ predicted observations well (Baker et al., 2018; Fan et
al., 2005; Hogrefe et al., 2009; Smyth et al., 2006), matching general trends in the observational data, and one saw an
overestimation of observational data (Hogrefe et al., 2015). Of the three aircraft studies, two saw significant underestimation
of $PM_{2.5}$ aloft (Baker et al., 2018; Chen et al., 2020), while one saw overestimation in some $PM_{2.5}$ compositional components
and underestimation in others (Yu et al., 2012).



Particle surface area specifically is not regulated as a criteria air pollutant as standards of measurement and air quality controls are determined on a mass per unit volume basis. However, particle surface area indirectly affects the concentration of $PM_{2.5}$ and $O_3$ as it can serve to regulate the lifetime of nitrogen oxides (Chang et al., 2011; Geyer and Stutz, 2004; Portmann et al.,

1996; Stadtler et al., 2018) and hydrogen oxides (George et al., 2013; Lakey et al., 2015; Martin et al., 2003; Thornton et al., 2008; Thornton and Abbatt, 2005), the production rate of secondary organic aerosol (Gaston et al., 2014), and new particle formation and growth rates, as preexisting aerosol surface area serves as a condensation sink for low volatility gas-phase compounds (Donahue et al., 2014; Trump et al., 2014). There are few reports of model-measurement comparisons of particle surface area, and those that have been reported in the literature have focused on comparisons of heavily spatially and temporally

averaged concentrations (e.g., field campaign averages). For example, Simon et al. compared ground-based aerosol surface area concentrations calculated in the CAMx model to measurements made aboard the *R.V. Ronald H. Brown* in the Gulf of Mexico and the Houston Ship Channel with two differential mobility particle sizers and an aerodynamic particle sizer (Bates et al., 2008; Simon et al., 2010). The results of these studies are given in Table 1. Model prediction of median $S_a$ in the Gulf of Mexico was similar to the measurement data ($S_{a,\text{mod}}/S_{a,\text{meas}} = 0.96$), with median values and ranges again shown in Table 1.

In comparison, model prediction of median $S_a$ in the Houston Ship Channel, where there is large spatial and temporal fluctuation in $S_a$, yielded $S_{a,\text{mod}}/S_{a,\text{meas}} = 1.6$. Modeled $S_a$ was also compared to measured $S_a$ aloft on two research flights from the TexAQS II/GoMACCS field study in September and October of 2006. The range of measured $S_a$ values (<600 μm$^2$/cm$^3$) matched well with the average values predicted in the CAMx model, though the maximum modeled values were much larger than those measured (4000-8000μm$^2$/cm$^3$ compared to <600 μm$^2$/cm$^3$), consistent with the *R.V. Ronald H. Brown* comparisons.

Overall, it should be noted that on a regional scale, modeled values agree well with measurements of aerosol $S_a$, however maximum modeled values were larger than those measured both for ground-based measurements and those aloft.





| Region | | Median (µm²/cm³) | Q1, Q3 (µm²/cm³) | Reference |
|---|---|---|---|---|
| Gulf of Mexico | Measured | 361 | 277, 398 | Simon et al. 2010 |
| | CAMx Model | 347 | 298, 394 | |
| Houston Ship Channel | Measured | 592 | 513, 800 | Simon et al. 2010 |
| | CAMx Model | 949 | 667, 1760 | |
| Northeast U.S. (surface - 0.1km) | Measured | 135 | 0.91, 2.23 | Jaeglé et al., 2018 |
| | GEOS-Chem Model | 169 | - | |
| Northeast U.S. (3.5 - 4.5 km) | Measured | 4.4 | 0.03, 0.17 | Jaeglé et al., 2018 |
| | GEOS-Chem Model | 3.8 | - | |

**Table 1: Comparison of model and measured aerosol surface area concentrations. Simon et al. (2010) compared CAMx model results with ground-based measurements in the Gulf of Mexico and the Houston Ship Channel. Jaeglé et al. (2018) compared measurements from the WINTER campaign and GEOS-Chem modeled data at the surface and aloft between 3.5 and 4.5km.**

More recently, modeled dry surface area concentrations were assessed over the northeast US during the 2015 Wintertime

INvestigation of Transport, Emissions, and Reactivity (WINTER) aircraft campaign (Jaeglé et al., 2018). While quantitative

assessment of aerosol surface area was not the focus of this study, dry aerosol $S_a$ was calculated by combining dry aerosol size

distribution observations from a Passive Cavity Aerosol Spectrometer Probe and Ultra-High Sensitivity Aerosol Spectrometer

and comparing to the GEOS-Chem chemical transport model. Two versions of the model, a reference and improved model

were compared to the observations within 13 altitude bins, ranging from surface to 4.5km, here we focus on the improved

model results. The GEOS-Chem model medians were encompassed by the observed interquartile ranges in each altitude bin.

The improved model showed excellent agreement with measurements when compiled over large spatial and temporal scales,

where $S_{a,\mathrm{mod}}/S_{a,\mathrm{meas}}$ was 1.25 and 0.68 for the surface and 4.5km comparisons, respectively.

Given the importance of accurate model representation of aerosol surface area to multiple atmospheric processes, and the

limited number of prior studies conducted in urban environments, we revisit this comparison using a regional air quality model aerosol and commensurate aircraft observations conducted in an urban environment.

## 2 Methods and Models

### 2.1 Aerosol Evaluation in the CMAQ Model

CMAQ simulations were performed as described by Abel et al. 2018, Abel et al. 2019, and Harkey et al. 2021, with carbon

bond 5 chemistry, anthropogenic emissions from the 2011 National Emissions Inventory, and input meteorology constrained to the North American Regional Reanalysis (NARR; Messinger et al., 2006). The CMAQ simulation utilized here employed CMAQ version 5.2.1 (Byun and Schere, 2006; Nolte et al., 2015), and was run with 25 vertical layers from the surface to 100 hPa, a $12 \times 12$km latitude and longitude grid, and hourly temporal resolution. The meteorology is from the Weather Research and Forecasting (WRF) version 3.2.1 (Skamarock et al., 2008), run including temperature, humidity, and wind from NARR.

Anthropogenic emissions and emissions from fires (both prescribed and not) were based on the 2011 National Emissions Inventory, with in-line estimates of NO and $NO_2$ produced by lightning, boundary conditions from the Model for Ozone and Related Chemical Tracers version 4, and biogenic emissions from WRF output in the Model of Emissions of Gases and Aerosols from Nature version 2.1 (Guenther et al., 2012). The model was run from 20 May through 31 August 2011, to include 11 days of spin-up (Harkey et al., 2021). The data set utilized in this analysis is only a subset of the model data set originally

run at UW-Madison.

CMAQ was also run for the time period of the 2015 WINTER field campaign for comparison of modelled and measured $N_2O_5$ uptake coefficients. This CMAQ simulation also employs input meteorology constrained to NARR, calculated using WRF version 3.8.1 (Skamarock et al., 2008). Anthropogenic emissions, emissions from fires, and boundary conditions from the

EPA Air QUAlity TimE Series (EQUATES) project (https://www.epa.gov/cmaq/equates). Biogenic emissions and lightning NOx emissions were both calculated in-line. This simulation employed CMAQ version 5.3.2 (Appel et al., 2021), with carbon bond 6 chemistry (Emery et al., 2015; Luecken et al., 2019). The WINTER-period simulation was run from 21 January through

16 March 2015, to include 11 days of spin-up, with hourly output on a 12 x 12 km grid and on 35 vertical levels from the surface to 100 hPa.


The CMAQ simulation for the 2011 DISCOVER-AQ period employed the "AERO6" aerosol module (Binkowski and Roselle, 2003; Carlton et al., 2010; Foley et al., 2010; Sonntag et al., 2014), where primary and secondary aerosols are characterized by bimodal lognormal size distributions, and the total size distribution is the sum of three aerosol size modes: Aitken, accumulation, and coarse modes with median diameters of 0.03, 0.3, and 6μm respectively. The CMAQ simulation for the 2015 WINTER period employed the "AERO7" aerosol module, which builds on the AERO6 module, with updates to aerosols formed by monoterpene oxidation, anthropogenic volatile organic compounds, and to aerosol liquid water (Pye et al., 2015; Pye et al., 2017; Qin et al., 2021; Xu et al., 2018). The CMAQ simulation for the DISCOVER-AQ period employed the default heterogeneous $N_2O_5$ uptake (Davis et al., 2008), while the simulation covering the WINTER period employed a $N_2O_5$ uptake modified per Bertram and Thornton (2009).


Due to the modality of the CMAQ representation of aerosols, we calculate each parameter relating to the aerosol data set separately for each mode, and these are then combined to result in a total value that can be directly compared to the DISCOVER-AQ observational data. The total CMAQ dry surface area is computed as the sum of the modal dry surface areas. The variable SRF is an output of CMAQ but is defined as $SRF = \pi\, M_2^d$. SRF is a modal variable like each moment, such that total surface area = SRFATKN + SRFACC +SRFCOR (dry surface area in the Aitken, accumulation, and coarse modes, respectively).

## 2.2 DISCOVER-AQ 2011 Campaign

Research flights conducted during the NASA DISCOVER-AQ (Deriving Information on Surface Conditions from COlumn and VERtically Resolved Observations Relevant to Air Quality) campaigns were designed to measure the vertical and spatial distribution of key air pollutants in urban environments, with a focus on connecting surface measurements with vertically integrating satellite observations. The first DISCOVER-AQ campaign, conducted aboard the NASA P-3B aircraft during July





2011, comprised of 14 science flights in the Baltimore and Washington, DC area (Crawford et al., 2014; Crawford and

Pickering, 2014; NASA, 2012). The 2011 DISCOVER-AQ campaign was the first of a series of flights with an objective of

narrowing the gap of satellite and observational data and air quality utilizing near-surface air pollution measurements. Science

flights concentrated on high time resolution measurements of atmospheric composition in the convective boundary layer. Here,

we focus on measurements of dry aerosol surface area concentration ($S_a$), determined from high time resolution (1Hz) size

distributions made using an Ultra-High Sensitivity Aerosol Spectrometer (Droplet Measurement Technologies, UHSAS)

integrating between $60 < d_p < 1000$ nm, which captures the peak of the surface area distribution, shown in Figure 1 below. The

UHSAS measures the particle size from optical light scattering, which was calibrated during DISCOVER-AQ using NIST-

traceable polystyrene latex spheres whose refractive index may differ slightly from that of real-world, aerosols that may result

in a slight under-sizing bias (Moore et al., 2021). the particle mobility size from 10-310 nm diameters was measured with a

TSI Scanning Mobility Particle Sizer (SMPS) with 45-second time resolution and the particle aerodynamic size from 500-

4000 nm diameters was measured with a TSI Aerodynamic Particle Sizer (APS) at 1 Hz. On the representative day shown in

Figure 1, SMPS data showed that approximately 5.8% of the surface area fell below the 60nm threshold of the UHSAS

measurement for an average surface area distribution, while the APS indicated that supermicron particles did not contribute to

the particle surface area. For simplicity, we choose to focus exclusively on the UHSAS size distribution data given their high

frequency and wide size range of particle diameter, but it is important to note that not all of the particle surface area is captured

by the UHSAS instrument. UHSAS data is available to the public at the NASA Langley Atmospheric Science's Data Center

and Distributed Active Archive Center (http://doi.org/10.5067/Aircraft/DISCOVER-AQ/Aerosol-TraceGas). In the following

analysis we utilize observations of nitric oxide (NO) and nitrogen dioxide ($NO_2$) measured with the NCAR four channel

chemiluminescence instrument (Ridley and Grahek, 1990) and carbon monoxide (CO) measured via Differential Absorption

CO Measurement (DACOM) (Sachse et al., 1987) to assess differences in modeled and measured aerosol surface area.

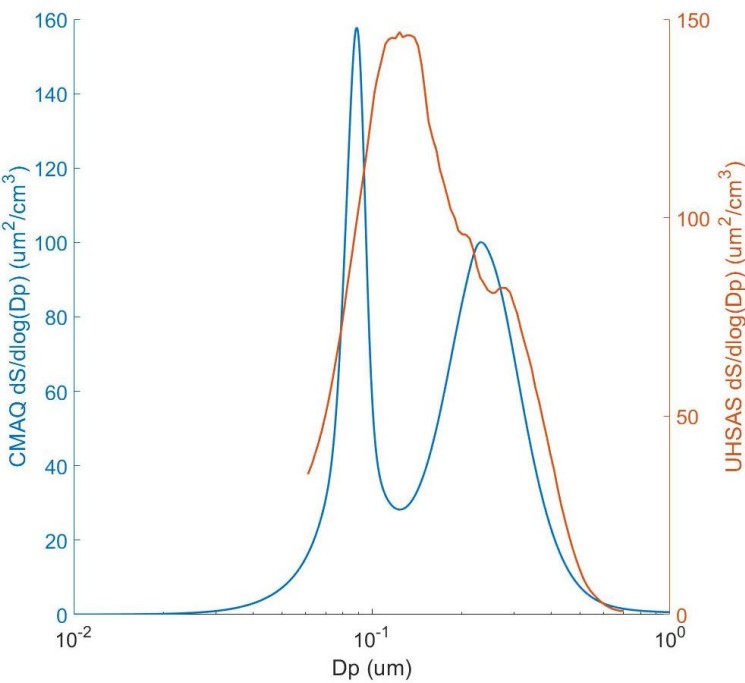


**Figure 1: Modeled (blue) and measured (orange) dry aerosol surface area distributions averaged over the DISCOVER-AQ sampling domain of Baltimore and Washington, DC for 1 July 2011.**

**2.3 Model-Measurement Comparison**

To compare measured and modeled $S_a$, we sample the hourly 12km × 12 km CMAQ output at the time and location of each

DISCOVER-AQ sampling point. The spatial resolution of CMAQ, relative the DISCOVER-AQ flight, is shown in Figure 2

for the 28 July 2011 research flight, where the color corresponds to the modeled surface-level dry $S_a$ (in $\mu m^2$ $cm^{-3}$) at noon

EST. Indexing and analysis of the two aforementioned data sets was completed in MATLAB. Each 1s data point from the

DISCOVER-AQ 2011 campaign was mapped to the nearest (as described below) 4D index (time of day, latitude, longitude,

and altitude) in the lower time and spatial resolution CMAQ model for direct comparison. The nearest 1hr averaged CMAQ

time point was selected based on the time window that encompassed the aircraft flight time, i.e. a flight time of 09:16:00 was

mapped to CMAQ time period of 9:00 – 9:59. The nearest 12km × 12km CMAQ grid box was also selected based on the grid

box that encompassed the aircraft location at the time of sampling. The nearest CMAQ altitude (or layer) was identified by

locating the aircraft height within one of the 25 indexes in which the altitude was encompassed. With all four CMAQ indexes

assigned to each data point, the 1s DISCOVER-AQ data set could be fully mapped and compared to that from the CMAQ





model. It should be noted that there are far more data points in the observed data than in the model due to resolution constraints, and thus the model is being oversampled. Each of the four indexes were concatenated together in the order defined in CMAQ, namely latitude, longitude, layer (altitude), and time. This process was utilized for each data point for an entire flight of DISCOVER-AQ and was then replicated for each subsequent flight. The result of this approach is shown in Figure 3, for the comparison of modeled and measured carbon monoxide (CO). The coefficient of determination ($r^2$) for the linear regression

of modeled vs measured CO concentration ($CO_{mod}$ / $CO_{meas}$) was 0.44 with a slope of $1.0499 \pm 0.0007$. The large variance highlights the spatial and temporal mismatch of model sampling and measurement, while the near unit correlation coefficient indicates that on average modeled and measured CO agree. This agreement implies that the model-measurement comparison of many well-understood parameters should be accurate and that there is not a fundamental issue in comparing modeled and measured data between the two datasets.

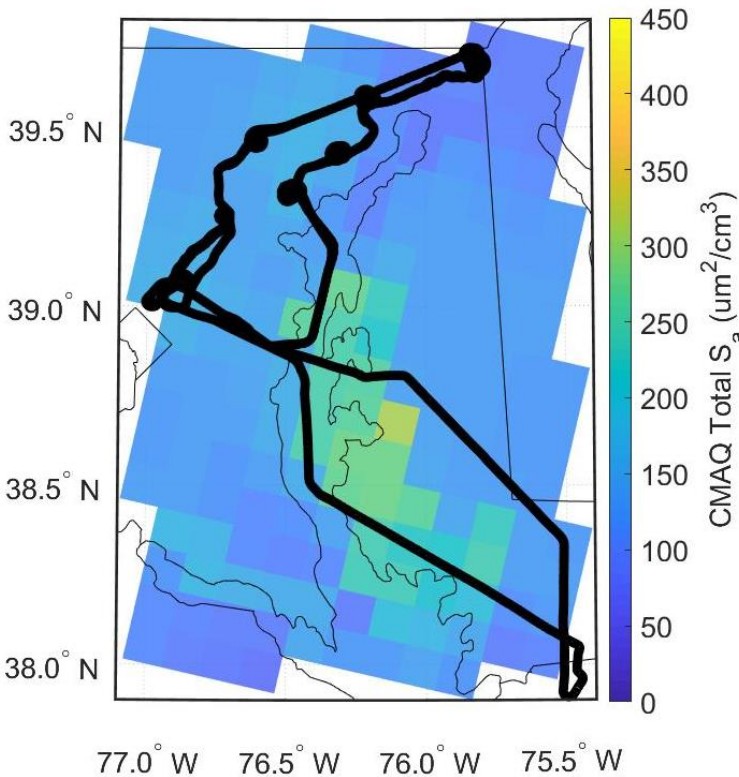


**Figure 2: Map of the Baltimore-Washington DC area representative of the July 2011 DISCOVER-AQ flights with an example flight path from 28 July 2011. The flight path is overlayed on the CMAQ model grid, showing the modeled surface area in each 12km × 12km grid box at noon EST (UTC 16).**




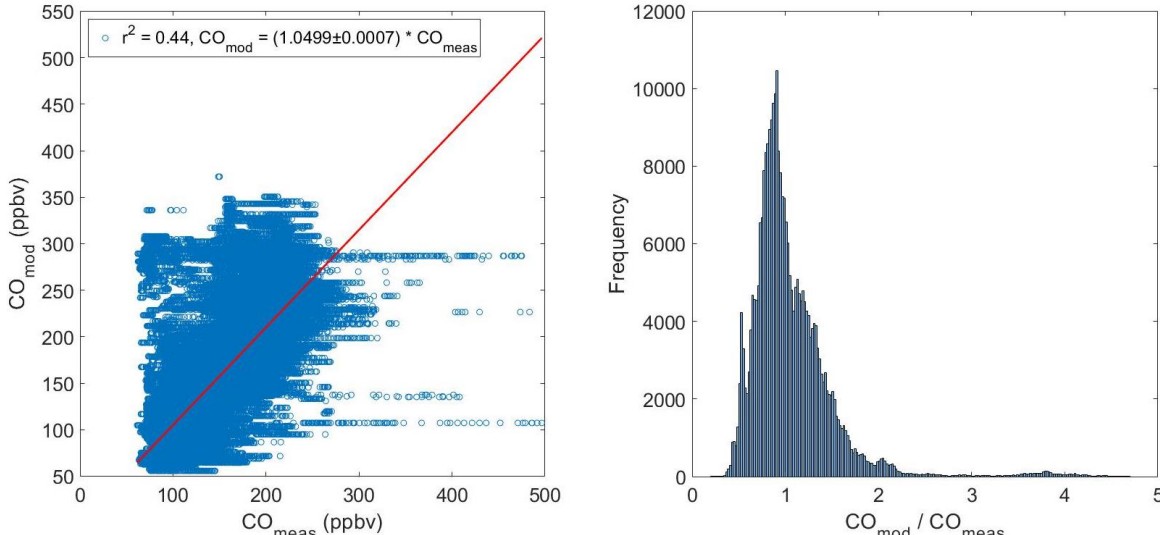

**Figure 3: a) Linear regression of modeled vs measured carbon monoxide (CO) mixing ratio in ppbv and b) histogram of the modeled-measured CO for the entire DISCOVER-AQ campaign.**

## 3 Results

### 3.1 Campaign Averaged Comparison of Aerosol Surface Area Concentrations

First, we assess general agreement between campaign averaged modeled and measured surface area concentrations. The observational data is from the 13 DISCOVER-AQ flights during July 2011 (1-29 July). The final research flight consisted of a dual highway leg conducted south along the Baltimore-Washington Parkway and north along I-95 at low altitude to compare the two roadways. Given the proximity to a large point source, this research flight was not included in the following analysis. The observational data from the UHSAS included number, surface area, and volume measurements, though the focus of this study is primarily surface area ($S_a$).

In this analysis, we compare dry surface area concentrations as the DISCOVER-AQ measurements were made dry and we do not have direct measurements of particle growth factors for comparison of wet $S_a$. However, it is important to note that $S_a$ used in E2 is the surface area concentration at ambient humidity and any uncertainty in modeled aerosol hygroscopicity will propagate to the aerosol surface area concentration used in E2. Figure 4 shows the campaign-averaged vertical profile of both





the measured dry UHSAS surface area ($S_{a,\text{meas}}$) as well as the modeled dry CMAQ surface area ($S_{a,\text{mod}}$), along with the interquartile ranges separated into 1km altitude bins. Since the UHSAS measurement frequency is 1 Hz and the CMAQ modeled data is at one-hour time resolution, and the model samples a full domain in 12km grid boxes compared to the smaller domain sampled by aircraft, there are many more measurement data points (N = 330204) than comparable modeled data points (N = 5196) over the course of the flight campaign. In Figure 4, the UHSAS measurements have been averaged to the spatial and temporal resolution of the model, such that the number of observational points is the same as the number of model points. The light gray error bars shown in Figure 4a reflect the standard deviation of the data from the mean at that point in time and space. It should be noted that the error bars on this dataset are large, due to the spatial and temporal mismatch between model and measurement in a highly heterogeneous sampling domain. For both model and measurement, the surface area increases towards the surface, as is to be expected, and decreases with altitude. The vertical profile is well captured by the CMAQ model, however, there is a larger range of measured surface area concentrations than is seen in the corresponding model altitude bin.

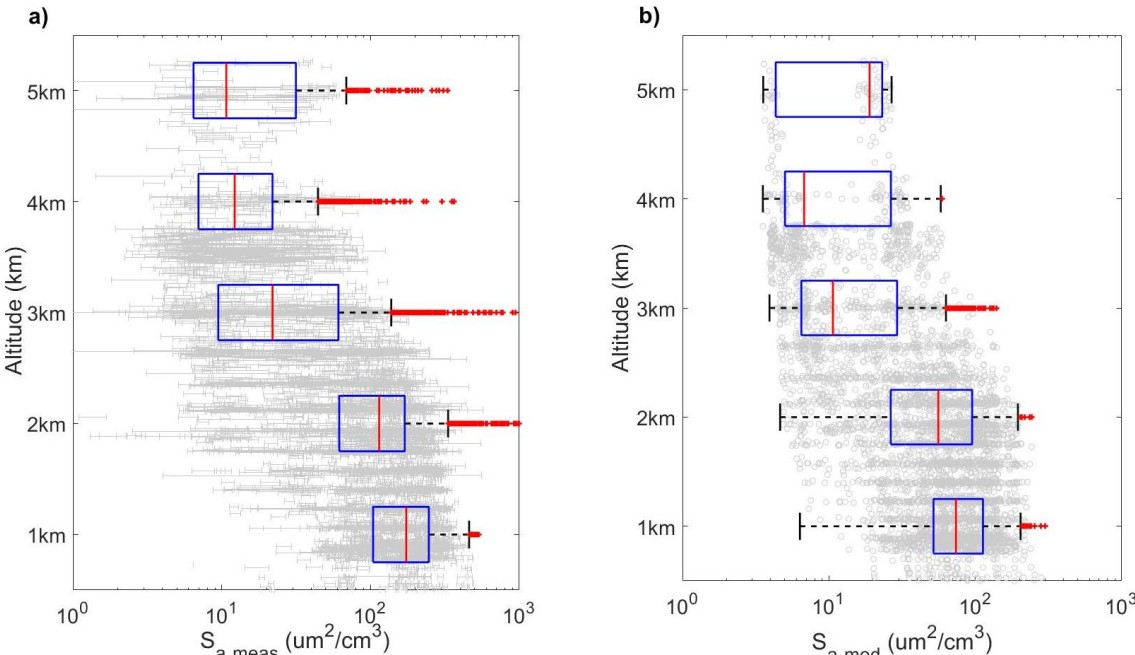

**Figure 4: Average vertical profile of a) measured DISCOVER-AQ aerosol surface area concentration ($S_{a,\text{meas}}$) and b) CMAQ aerosol surface area concentration ($S_{a,\text{mod}}$) over the entirety of the DISCOVER-AQ campaign. Each measured point is an average of the points included in that 4D index corresponding to CMAQ. The overlayed boxplots show the median (red line within blue box), and interquartile ranges (blue box with the 25th percentile at the left end and 75th percentile at the right end) in 1km altitude bins. The labels on the altitude axis lie at the midpoint of the 1km altitude bin, and red crosses indicate outliers from the majority of the dataset at that altitude portion.**





As shown in Figure 4, measured $S_a$ is on average larger than modeled $S_a$, particularly at low altitude, where the ratio of the median, modeled $S_a$ to measured $S_a$ near the surface ($z < 1$ km) is 0.47. This contrasts with what has been reported previously in the literature. For example, both Jaeglé et al and Simon et al. found that the median modeled near-surface $S_a$ was consistently larger than measured $S_a$ ($S_{a,mod}/S_{a,meas} = 1.04 – 1.6$) (Simon et al., 2010).

The comparison between modeled and measured $S_a$ is also shown with histograms in Figure 5 for all altitudes (5a,b) and for the surface level measurements (0-1km; 5c,d). While the number of points is not consistent between the model and measurement datasets due to the 12km grid box constraint and time frequency in CMAQ, differences in the range in surface area concentrations are observed. Measured surface area concentrations range between 0 - $1.87 \times 10^3$ μm²/cm³ with the vast majority of data below 420 μm²/cm³, while modeled $S_a$ ranges between 0-300μm²/cm³.

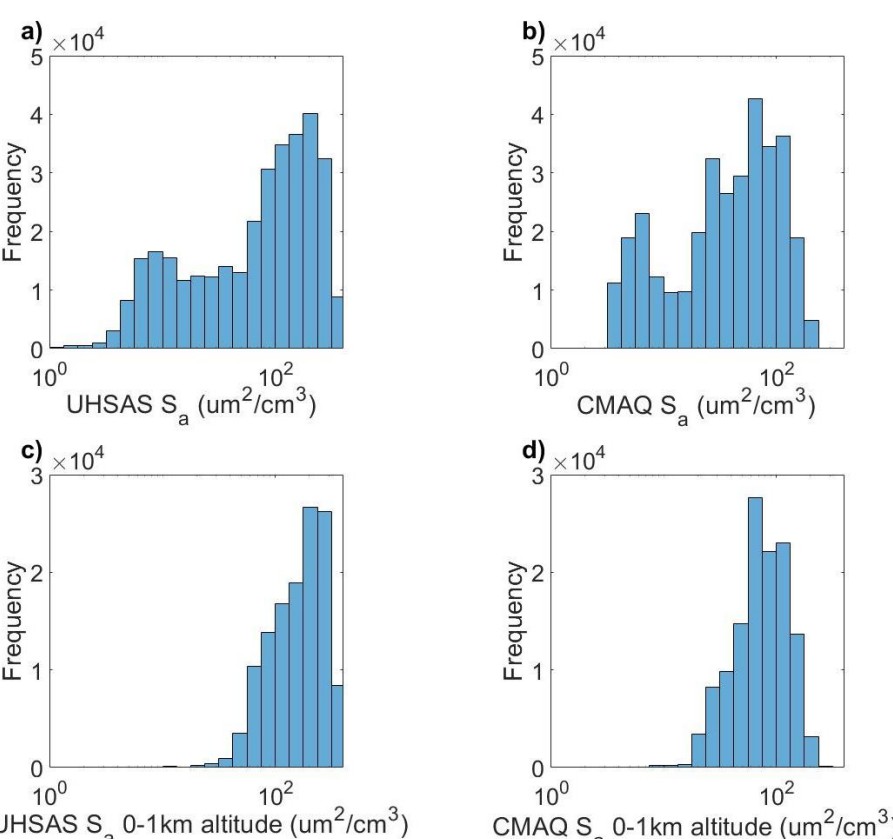


**Figure 5: Histograms of a) measured aerosol surface area concentration (DISCOVER-AQ), b) modeled aerosol surface area concentration (CMAQ), c) measured aerosol surface area in the 0 to 1km altitude bin, and d) modeled aerosol surface area in the 0 to 1km altitude bin over the entirety of the DISCOVER-AQ campaign.**





## 3.2 Direct Model-Measurement Comparison

A linear regression of CMAQ modeled dry aerosol surface area concentration and measured aerosol surface area concentration

is shown in Figure 6. The measured data has been averaged to the space and time domain of CMAQ (latitude, longitude,

altitude, and time). The coefficient of determination ($r^2$) for the linear regression of modeled and measured $S_a$ was 0.52 with a

slope of $0.437 \pm 0.004$, indicating that the measured $S_a$ is on average twice that of the model value.

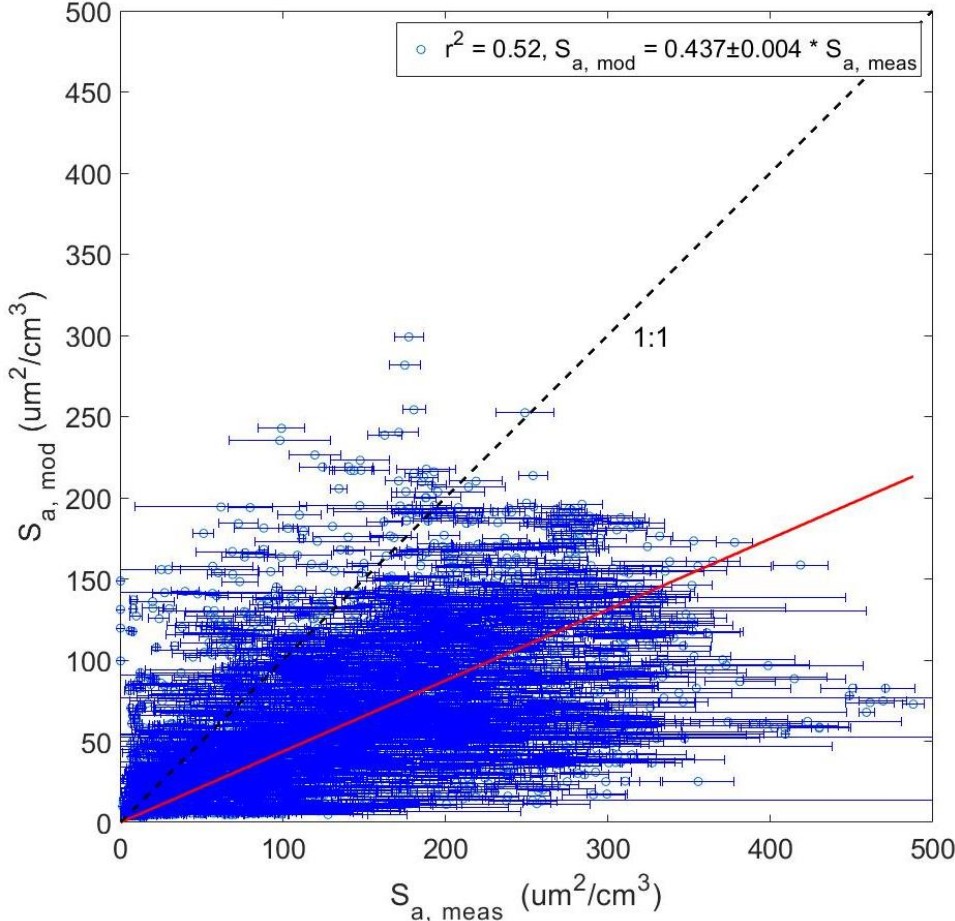

**Figure 6: Comparison of $S_{a,mod}$ and $S_{a,meas}$ over the full DISCOVER-AQ campaign with measurement data averaged to the corresponding model latitude, longitude, altitude, and time point.**

The histogram of the surface area ratio ($S_{a,mod}/S_{a,meas}$) throughout the campaign in Figure 7 shows that the model underpredicts

the measured surface area ratio in 81% of the comparison points. The model underpredicts $S_a$ by a factor of two 44% of the

time. In the following section, we explore potential causes for model-measurement disagreement, including model-





measurement spatial and temporal differences, the spatial distribution of primary emissions, and/or treatment of secondary

aerosol formation.

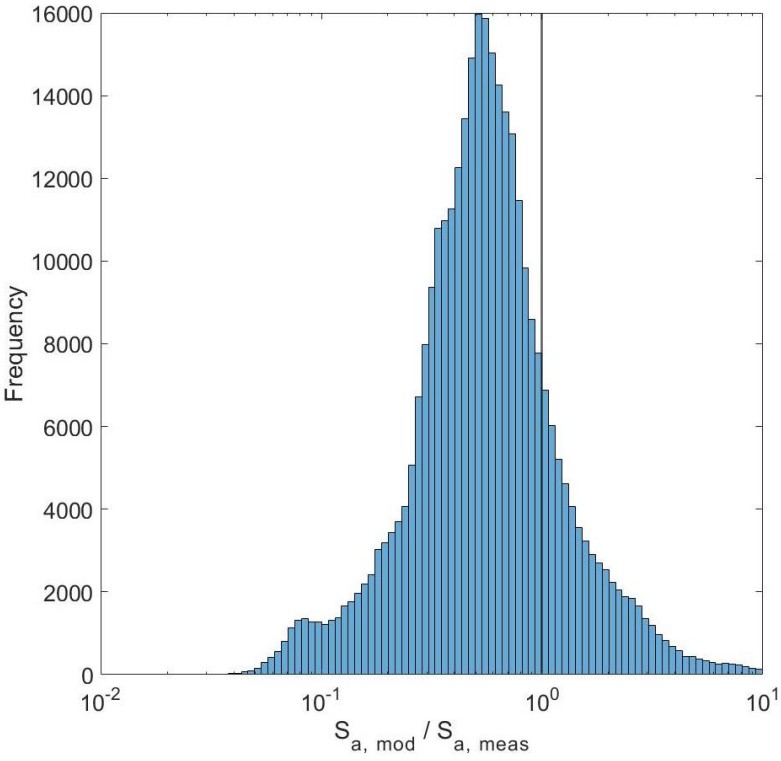

**Figure 7: Histogram of the ratio of model to measured aerosol surface area concentration ($S_{a,\mathrm{mod}}/S_{a,\mathrm{meas}}$) over the full DISCOVER-AQ campaign.**


## 4 Discussion

In the following section, we explore the source of model-measurement discrepancy in $S_a$ discussed in section 3. We begin by

investigating the dependence of $S_{a,\mathrm{mod}}/S_{a,\mathrm{meas}}$ on altitude and proximity to primary aerosol sources. We then investigate the

role of temporal and spatial resolution as CMAQ has a much coarser resolution, both spatially and temporally, than the

measured data. Finally, we investigate the possibility of impacts on $S_{a,\mathrm{mod}}/S_{a,\mathrm{meas}}$ from anthropogenic and biogenic indicators

as they are tied to aerosol emissions.





## 4.1 Dependence of $S_{a,mod}/S_{a,meas}$ on Altitude

Given the strong dependence of $S_a$ on altitude as shown in Figure 4, we first explore if part of the variance in $S_{a,\mathrm{mod}}/S_{a,\mathrm{meas}}$

shown in Figure 6 can be explained by altitude. In Figure 8, we show $S_{a,\mathrm{mod}}/S_{a,\mathrm{meas}}$ as a function of altitude. As shown, there is

an altitude dependence in $S_{a,\mathrm{mod}}/S_{a,\mathrm{meas}}$, where the mean, median, and interquartile range (25th to 75th percentile) are given in

Table 2 for the 1km altitude bins from 0-5km. Model-measurement discrepancy in $S_a$ is largest at low altitude, where particle

number concentrations are highest, proximity to particle sources is close, and heterogeneity in particle number concentrations

are largest.

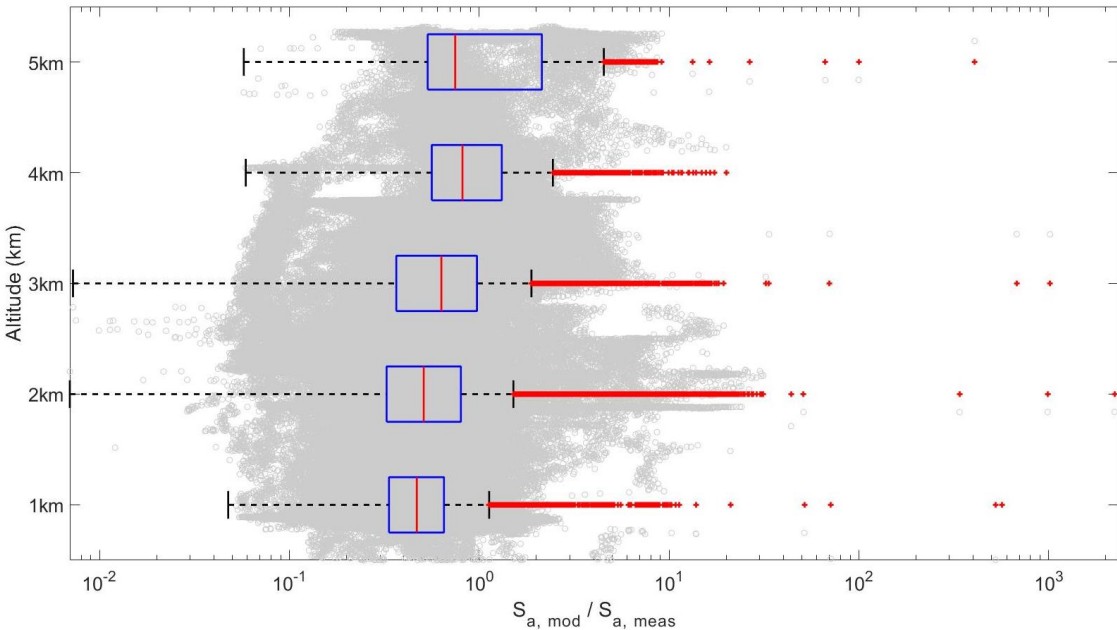

**Figure 8: Ratio of modeled to measured aerosol surface area concentration ($S_{a,\mathrm{mod}}/S_{a,\mathrm{meas}}$) including median (red line within blue box), and interquartile ranges (blue box with the 25th percentile at the left end and 75th percentile at the right end) in 1km altitude bins. The red crosses outside of the bounds of the plot note outliers. The labels on the altitude axis lie at the midpoint of the 1km altitude bin.**

| | 0-1km | 1-2km | 2-3km | 3-4km | 4-5km |
|---|---|---|---|---|---|
| $S_{a,mod}/S_{a,\mathrm{meas}}$ (mean) | 0.56 | 0.97 | 0.89 | 1.05 | 1.42 |
| $S_{a,mod}/S_{a,\mathrm{meas}}$ (median) | 0.47 | 0.51 | 0.63 | 0.82 | 0.75 |
| $S_{a,mod}/S_{a,\mathrm{meas}}$ (interquartile range) | 0.33-0.65 | 0.33-0.80 | 0.37-0.97 | 0.56-1.31 | 0.53-2.14 |

**Table 2: Mean, median, and interquartile range (range of 25th to 75th percentile) surface area ratio ($S_{a,\mathrm{mod}}/S_{a,\mathrm{meas}}$) for each 1km altitude bin from 0-5km.**





## 4.2 Dependence of $S_{a,mod}/S_{a,meas}$ on Spatial and Temporal Resolution

The 12 km ×12 km spatial resolution and 1 hour temporal resolution of CMAQ is significantly larger and longer than the spatial and temporal resolution of the aircraft data, resulting in an inherent contrast in resolution between model and

measurement that may play a role in the variance in $S_{a,mod}/S_{a,meas}$. Within any individual 12 km ×12 km model pixel, in the Baltimore-Washington sampling area, there is heterogeneity in $S_a$ as shown in Figure 9. Sub grid scale variability in $S_a$ would lead to increased variance in $S_{a,mod}/S_{a,meas}$, but likely with a mean and median close to 1 if the domain sampling was not biased, comparable to what is observed in the CO comparison (Figure 3), where the histogram of the CO data showcases a clear center around 1, with very few data points beyond a $CO_{mod}/CO_{meas}$ value of 2.

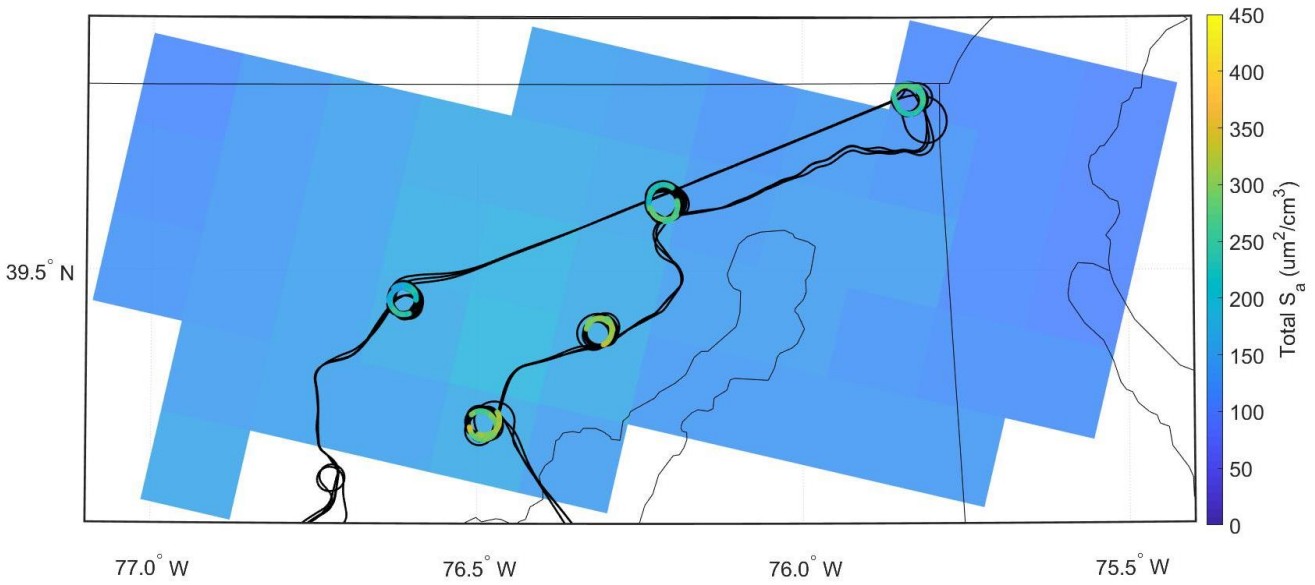


**Figure 9: Flight path from 28 July with overlayed UHSAS $S_a$ data within the 10th layer of CMAQ altitude (~850-1000m) and gridded CMAQ $S_a$ from layer 10 in the background. The CMAQ data is specifically at noon EST (UTC 16).**

To investigate the discrepancy more quantitatively, we compare the probability density functions (PDF) of the model-to-

measured CO, $NO_x$, particle number concentration and particle surface area concentrations. We use the PDF to characterize the population of data based on the standard deviation and mean, which provides a quantitative and comparable assessment of the variability in the comparison. Assuming that the research flights sampled the CMAQ model domain in an unbiased way (i.e. flights did not target or avoid point sources) we would expect that the PDFs of the model-to-measured ratio in CO, $NO_x$,



number concentration, and surface area concentration would all center at 1 (or $\log_{10}(1) = 0$ as shown in Figure 10) and the

standard deviation of the distribution ($\sigma$) would reflect heterogeneity in the scalar concentration at scales smaller than the

model spatial or temporal domain. The histogram, PDF, and cumulative distribution function (CDF) for $\log_{10}(S_{a,\,mod.}/S_{a,\,meas.})$

is shown in Figure 10. The PDF of the histogram of $\log_{10}(S_{a,\,mod.}/S_{a,\,meas.})$ has a mean ($\mu$) of -0.26 and standard deviation ($\sigma$) of

0.34.

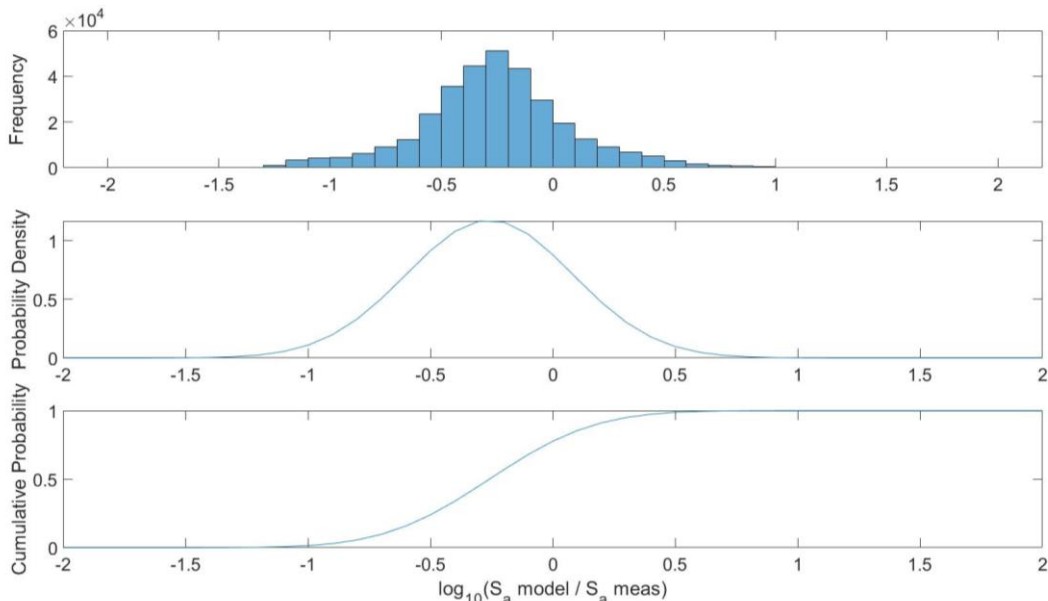

**Figure 10: Normalized histogram, probability density function (PDF), and cumulative distribution function (CDF) for the model-to-measuremed aerosol surface area ($S_{a,\,mod.}/S_{a,\,meas.}$). The histogram and PDF serve to indicate the median and spread of the dataset, while the CDF indicates the percentage of data encompassed at a certain data threshold.**

Comparison of the peak and width of the PDF of the model-to-measuered ratios of CO, particle number concentration ($N$), and

NO$_x$ provides an objective measure for assessing the impact of spatial and temporal resolution on the comparison. As shown

in Figure 11 and Table 3, the mean of each PDF is -0.0029, 0.047, and -0.14 for CO, $N$, and NO$_x$. Each of these values is

significantly closer to 0 than that measured for $S_a$ (-0.26), suggesting that the methodology for assessing model-measurement

agreement should not be significantly impacted by model resolution, especially given the large range in atmospheric lifetimes

for CO, $N$, and NO$_x$. Interestingly, the mean $N_{mod.}/N_{meas}$ is close to 1 ($10^{0.047} = 1.11$), where a value closer to 1 indicates

agreement between model and measurement and that closer to zero indicates a large discrepancy between datasets. The mean

$N_{mod.}/N_{meas}$ is significantly different than that observed for $S_{a,\,mod}/S_{a,meas}$ ($10^{-0.26} = 0.55$), perhaps suggesting that the model-





measurement disagreement is related to the shape of the size distribution either due to the *a priori* emissions size distribution

or secondary aerosol processes.

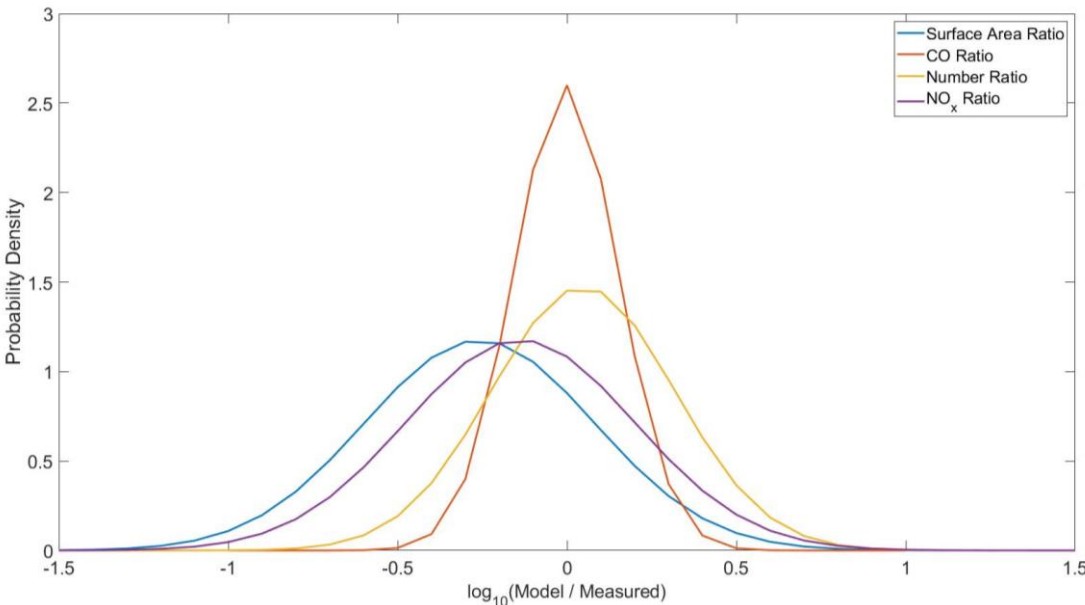

**Figure 11: Normalized probability density functions for the $\log_{10}$ of the model to measurement ratio in particle number concentration (yellow), carbon monoxide (red), NO$_x$ (purple), and particle surface area concentration (blue).**

|  | mean (μ) | standard deviation (σ) |
|---|---|---|
| $\log_{10}(S_{a,mod} / S_{a,\text{meas}})$ | -0.26 | 0.34 |
| $\log_{10}(N_{,mod} / N_{,\text{meas}})$ | 0.047 | 0.27 |
| $\log_{10}([CO]_{mod} / [CO]_{\text{meas}})$ | -0.0029 | 0.15 |
| $\log_{10}([NO_x]_{mod} / [NO_x]_{\text{meas}})$ | -0.14 | 0.34 |

**Table 3: Probability Density Function (PDF) fit parameters for the distributions shown in Figure 11.**


Also shown in Figure 11 and Table 3 the standard deviation (σ) of the PDF for the CO, $N$, and NO$_x$ model-to-measurement

ratios, is 0.15, 0.27, and 0.34 respectively. The standard deviation of the PDF of $[CO]_{mod}/[CO]_{\text{meas}}$ is the narrowest, likely

reflecting the longer lifetime of CO and a damping of sub grid scale variability of CO in each pixel. The width of the $N$ and $S_a$



ratio distributions are comparable, again highlighting that the deviation of $S_{a,\mathrm{mod}}/S_{a,\mathrm{meas}}$ from 1 may reflect differences in the
model-measured aerosol size distribution (as shown in Figure 1). Collectively, this analysis suggests that there is not a
significant bias on average in the methodology based on model resolution and that the apparent differences in the number and
surface area model-measurement ratios are most likely driven by the shape of the underlying aerosol size distribution.

It is interesting to note that the model-measurement agreement in particle number concentrations is significantly better than
that of particle surface area concentrations, implying that the differences in $S_a$ may be related to the shape of the aerosol size
distribution. There have been numerous analyses of model-measurement comparison of the aerosol number concentration and
size distributions specific to CMAQ (Elleman and Covert, 2009, 2010; Kelly et al., 2011; Zhang et al., 2010b). Elleman and
Covert compared the 4km CMAQ v4.4 model's size distributions to measurement data from the 2001 Pacific Northwest and
Pacific field campaigns. The Pacific Northwest field campaign (PNW2001) was conducted in August 2001 with both airborne
and ground-based measurements of pollution in the Puget Sound urban area around Seattle, Washington, and included
northwest Oregon, western Washington, and southwest British Columbia. PNW2001 was conducted to complement that of the
Pacific 2001 field campaign, which was a major regional air pollution study in the Lower Fraser Valley of metropolitan
Vancouver, British Columbia focusing on ground-based observations, conducted from 10 August to 2 September 2001.
Analyses of these two campaigns and model predictions found that CMAQ underpredicted airborne particle number
concentrations by a factor of 10-100 and was least accurate in the smallest size mode: the Aitken mode (Elleman and Covert,
2009). The underprediction was consistent between measurement studies and did not depend on time and location. Zhang et
al compared CMAQ v4.4 to the 1999 Southern Oxidants Study and corroborated the findings of Elleman and Covert, that the
Aitken mode was significantly underpredicted in total number concentration (varying by up to 3 orders of magnitude), yielding
an overall underprediction of $PM_{2.5}$ in Atlanta (Zhang et al., 2010b).


In a follow-up analysis, Elleman and Covert used updated emissions size distributions to compare a summer 2001 case study
comprising data from a period of August 2001 with airborne and surface measurements from Pacific 2001 and PNW2001, as
was used in the original base case to CMAQ (Elleman and Covert, 2010). CMAQ still underpredicted the observable aerosol

number concentrations by about one order of magnitude with updated emission size distributions, which was an improvement
from the 1-2 orders of magnitude previously but pointed to issues within the model's prediction of aerosol number. Kelly et
al. then utilized the updated emissions size distributions from Elleman and Covert as well as the original distributions with in
CMAQ to compare to the 1998 California Regional $PM_{10}/PM_{2.5}$ Air Quality Study (CRPAQS) (Kelly et al., 2011). It was noted
that the simulated number size distributions from the improved emission simulation were about 20% lower than the
observations, while the standard-emission simulation was about a factor of 5 lower than the observations, confirming that the
updated emissions improved model-measurement agreement. The observed shape of the distributions also better matched the
updated emissions simulations. The improvement in model-measurement agreement showcases the necessity for accurate size
distributions and emissions within CMAQ and the impact on $S_a$ data.

### 4.3 Dependence of $S_{a,mod}/S_{a,meas}$ on Secondary Aerosol Production

Two potential reasons for the discrepancy between mean $S_{a,\mathrm{mod.}}/S_{a,\mathrm{meas}}$ (0.55) and mean $N_{\mathrm{mod.}}/N_{\mathrm{meas}}$ (1.11) are: 1) uncertainty
in the size distribution of primary aerosol particles, and 2) uncertainty in secondary aerosol production (i.e. the condensation
of low volatility material to existing aerosol particles). To investigate these two potential sources, we investigate the response
of $S_{a,\mathrm{mod.}}/S_{a,\mathrm{meas}}$ to photochemical age. We start by looking at the response of $S_{a,\mathrm{mod.}}/S_{a,\mathrm{meas}}$ to the $NO_x/HNO_3$ ratio (Figure 12),
where high $NO_x/HNO_3$ in this sampling region is indicative of air masses near an anthropogenic source, similar to that of a
$NO_x/NO_y$ clock (Kleinman et al., 2008; Pan et al., 2015; Tie et al., 2009). If the aerosol surface area of primary emissions is
underestimated in the model, we would expect $S_{a,\mathrm{mod.}}/S_{a,\mathrm{meas}}$ to be biased low at high $NO_x/HNO_3$. If the condensation rate of
low volatility anthropogenic species is underestimated in the model, we would expect $S_{a,\mathrm{mod}}/S_{a,\mathrm{meas}}$ to decrease with a
decreasing $NO_x/HNO_3$ ratio as the airmass ages. As shown in Figure 12, $S_{a,\mathrm{mod.}}/S_{a,\mathrm{meas}}$ is remarkably constant over a wide span
of $NO_x/HNO_3$ ratios (0.5-10), before tending to larger values at low $NO_x/HNO_3$. This trend is also seen in the dependence of
$N_{\mathrm{,mod.}}/N_{\mathrm{,meas}}$ on $NO_x/HNO_3$, suggesting a potential discrepancy in the model-measured lifetime of aerosol or treatment of
background aerosol particles in the region. This trend suggests that an underestimate in the condensation of low-volatility gas-
phase compounds of anthropogenic origin are not a significant driver of model-measurement discrepancy in $S_a$. Rather, the





persistent underestimate of $S_a$ in the model at high $NO_x/HNO_3$ points to uncertainty in the size distribution of primary emissions

or secondary aerosol formed at the early stages of oxidation.

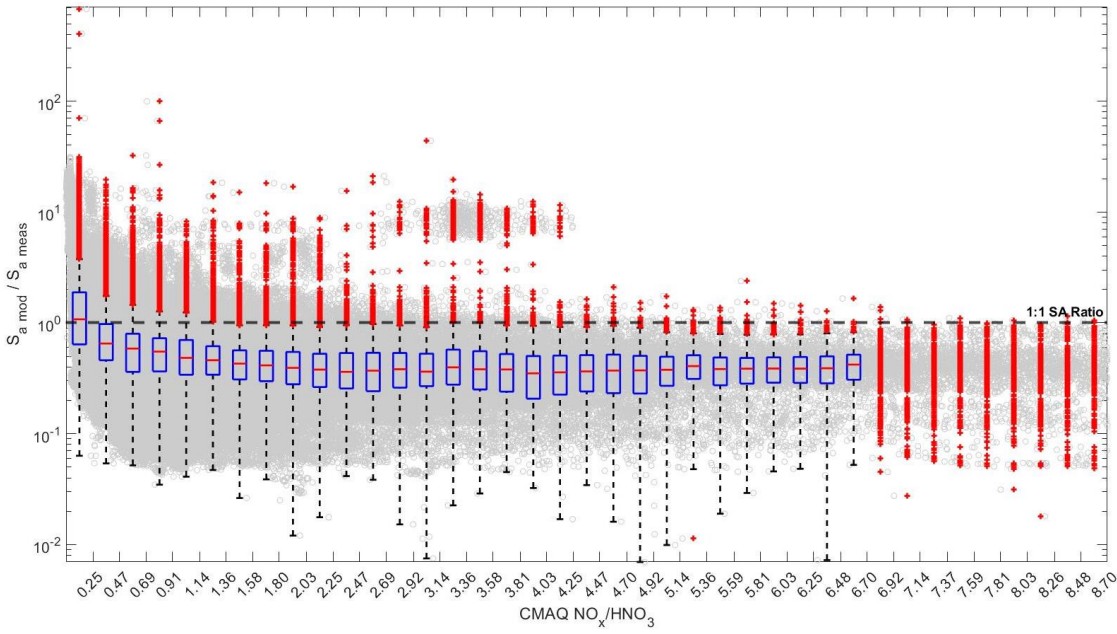


**Figure 12: Interquartile ranges in $S_{a,\text{mod.}}/S_{a,\text{meas}}$ as a function of the modeled (CMAQ) $NO_x/HNO_3$ concentration ratio.**

To address secondary aerosol formation more generally, we also assessed the response of $S_{a,\text{mod.}}/S_{a,\text{meas}}$ to temperature as

equilibrium partitioning in the gas-phase based on temperature and RH is a primary driver of secondary aerosol formation. No

statistically significant trend in $S_{a,\text{mod.}}/S_{a,\text{meas}}$ was observed over the range of temperatures observed during DISCOVER-AQ.

To further investigate secondary aerosol formation as a factor in driving the discrepancy in modeled $S_a$, we assess the response

of $S_{a,\text{mod.}}/S_{a,\text{meas}}$ to isoprene oxidation products in the aerosol phase as an example of biogenic VOC oxidation. As shown in

Figure 13, there does not appear to be a trend with concentration of isoprene SOA. Though we cannot test for all biogenic

oxidation products, the lack of a trend with isoprene SOA in the aerosol phase may mean that the discrepancy in $S_{a,\text{mod.}}/S_{a,\text{meas}}$

is not biogenic in nature.





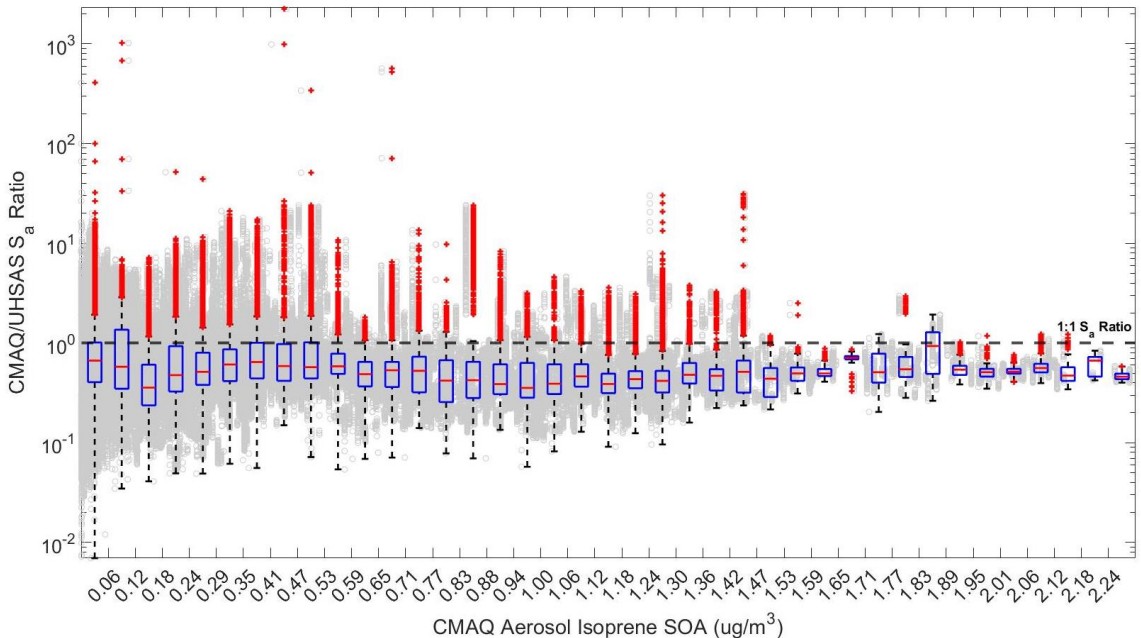

**Figure 13: Interquartile ranges in $S_{a,\text{mod.}}/S_{a,\text{meas}}$ as a function of the modeled (CMAQ) isoprene SOA concentration.**

## 5 Implications for the Treatment of Heterogeneous Reactions in Air Quality Models

As shown in E2, the rate constant for the heterogeneous loss of gas-phase compounds to aerosol ($k_{het}$) is linearly dependent on

both aerosol surface area concentration ($S_a$) and the reactive uptake coefficient ($\gamma$). In section 3, we showed that the average

$S_{a,\text{mod.}}/S_{a,\text{meas}}$, determined from the regression of the average model and measurement $S_a$ was 0.437, which would result in

approximately a factor of 2 underestimate in $k_{het}$. A similar underestimation has been seen previously in select ground (Ghim

et al., 2017; Liu and Zhang, 2011; Prank et al., 2016; Wang et al., 2021; Yu et al., 2008a, 2012, 2008b; Zhang et al., 2019,

2006, 2010c) and aircraft-based (Baker et al., 2018; Chen et al., 2020) studies of CMAQ prediction of $PM_{2.5}$, which may point

to a larger issue in model representation of particle mass. For some heterogeneous reactions, where the reactive uptake

coefficients are well parameterized in model (e.g., extremely low volatility species) uncertainty in $S_a$ likely determines

uncertainty in $k_{het}$. To assess the dominant source of uncertainty in model derive $k_{het}$, we focus on the $N_2O_5$ system as an

example. Recently, McDuffie et al. (2018) assessed the accuracy of model parameterizations of $\gamma(N_2O_5)$ using ambient

observations from the WINTER campaign. In Figure 14a, we show the histogram and PDF of the ratio of $\gamma(N_2O_5)_{mod}$ calculated

in CMAQ using the Bertram and Thornton parameterization for the WINTER campaign, compared with $\gamma(N_2O_5)_{meas}$, which





was determined in McDuffie et al. from an observationally constrained analysis of flight data (McDuffie et al., 2018). The

PDF of the directly compared model-measurement ratio is centered above zero ($\mu$ = 0.22 or $\gamma(N_2O_5)_{mod}/\gamma(N_2O_5)_{meas}$ = 1.65)

(Figure 14, Table 3). Interestingly, since $k_{het}(N_2O_5)$ is proportional to the product of $S_a$ and $\gamma(N_2O_5)$, the underestimate in model

$S_a$, which we assume would be consistent for WINTER, is compensated by an overestimate in $\gamma(N_2O_5)$ in the mean state. While

the width of the PDF of $\log_{10}(\gamma(N_2O_5)_{mod.}/\gamma(N_2O_5)_{meas.})$ for WINTER is similar to that seen for $S_a$ for DISCOVER-AQ, it should

be noted that neither the histogram of the $\gamma(N_2O_5)$ ratio or the $S_a$ ratio is easily fit to a gaussian peak shape. As shown in Figure

14, the histogram of the $\log_{10}(\gamma(N_2O_5)_{mod.}/\gamma(N_2O_5)_{meas.})$ for WINTER has a broader range of values than that of $S_a$ in this study.

Collectively, this analysis highlights that while model uncertainty in $k_{het}(N_2O_5)$ is largely a function of quality of the $\gamma(N_2O_5)$

parameterization, future improvements in modeled surface area concentrations, particularly in urban environments, will also

result in more accurate representations of heterogeneous chemical reactions.

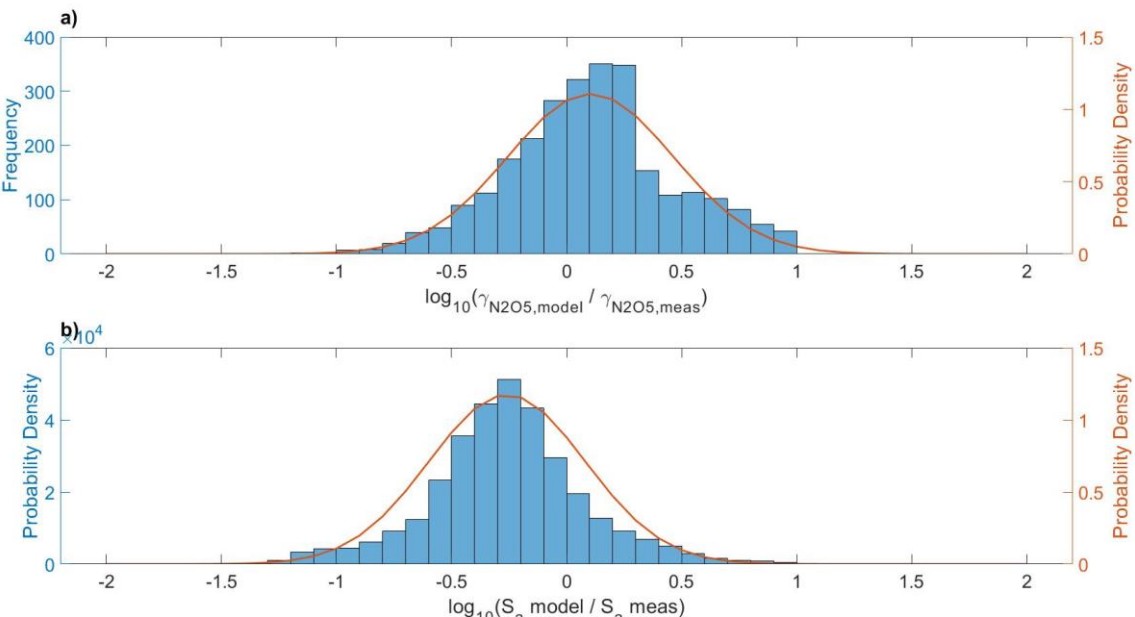

**Figure 14: Normalized probability density functions for the $\log_{10}$ of the model to measurement ratio in (a) $\gamma(N_2O_5)$ from the WINTER campaign and (b) particle surface area concentration, $S_a$ from this study.**

**Data Availability**

Data from the 2011 DISCOVER-AQ Campaign is publicly available at http://doi.org/10.5067/Aircraft/DISCOVER-AQ/Aerosol-TraceGas CMAQ output data will be made available on the University of Wisconsin – Madison MINDS database at https://minds.wisconsin.edu/handle/1793/76304.

**Author Contributions**

Monica Harkey, Alicia Hoffman, and Tracey Holloway ran the CMAQ model and assisted with analysis of model outputs. Rachel Bergin analyzed the model and measurement data. Rachel Bergin and Timothy Bertram wrote the paper. All authors reviewed and edited the paper.

**Competing Interests**

At least one of the (co-)authors is a member of the editorial board of Atmospheric Chemistry and Physics. The authors have no other competing interests to declare.

**Acknowledgements**

We would like to thank the DISCOVER-AQ NASA Langley Aerosol Research Group Experiment (LARGE) research team, including Richard Moore, Bruce Anderson, Andreas Beyersdorf, Luke Ziemba, Lee Thornhill, and Edward Winstead for the use of their data in this work. We would also like to thank Heather Simon, Lyatt Jaeglé, for their data contributions, and Erin McDuffie and Steve Brown specifically for the use of the WINTER data in this work.

**Financial Support**

This research has been supported by EPA project number R840006 and the NOAA Climate Program Office's Atmospheric Chemistry, Carbon Cycle, and Climate program (NA18OAR4310109). The NASA LARGE research team was supported by funding from the NASA Earth Venture Suborbital-1 Program.





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
