# Peer review of "Observation-Based Constraints on Modeled Aerosol Surface Area: Implications for Heterogeneous Chemistry"

_Atmospheric Chemistry and Physics, 2022_

## Author Comment (AC2)

We thank both anonymous reviewers for their helpful suggestions that we think make the revised manuscript a strong, more complete version of the original submission. In what follows we respond to each comment individually (in blue) and note where in the manuscript we have made revisions.

Responses to comments from Reviewer #1:

1. Particle surface area and number concentrations depend strongly on the size distributions of primary particles emitted (especially in the urban region like this study). It is unclear in the manuscript how primary particles of different sources (industry, residential, traffic, etc.) are parameterized in the model. This information is necessary and a table summarizing these will be helpful.

We agree, and the results presented in this paper suggest that the magnitude and size distribution of primary emissions are a leading cause for the differences in model-measured aerosol surface area. The scope of this paper was to assess the state of model-measurement agreement in particle surface area to determine how well regional models perform with respect to this important scalar in heterogeneous kinetics.

In CMAQ, primary particle emissions are given in units of mass/second, these are parameterized to $1^{st}$ and $2^{nd}$ moment emission rates following Binkowski and Roselle (2003). We are not able to "tag" primary emissions from different sectors post-emission; all primary emissions are parameterized the same way regardless of source. In the version of CMAQ used here, the emissions inventory for primary aerosol particles is the 2011 EPA National Emissions Inventory (https://www.epa.gov/air-emissions-modeling/2011-version-62-platform) split into categories including dust, agricultural, agricultural fire (prescribed burns), rail, marine, nonpoint, nonroad, nonpoint oil and gas, onroad, point oil and gas, point electrical generating units, residential wood combustion, point sources not associated with oil/gas/electricity, and estimates of point source and area emissions from outside the US (Canada, Mexico). The sources are speciated by location across the US and input into CMAQ.

The following text has been added to the manuscript at line 99, "Primary aerosol emission rates are sourced from the 2011 EPA National Emissions Inventory and split into sources by location. The size distribution of primary aerosol emissions and mixing into the model grid cell are input in CMAQ the same way regardless of source."

2. Particle size distributions (and thus Sa) are also affected by new particle formation (or nucleation) and growth, especially at higher altitudes and in areas away from the urban center. The authors shall discuss how new particle formation was treated in the model and how the process might have affected the results.

The data in this manuscript was run using version 5.2.1 of CMAQ. The default new particle formation nucleation parameterization in CMAQ version 4.3 and newer is based on a classical binary homogeneous nucleation which simulates kinetics and accounts for hydration (Kulmala et al., 1998). Only sulfuric acid is assumed to form new particles.

Regarding aerosol growth, the model assumes that organics do not influence aerosol water content (Binkowski and Roselle, 2003). The growth mechanism is described therein. The representation of SOA was updated in CMAQ version 4.7 to include SOA formation via gas-phase oxidation, aqueous-phase oxidation, and aerosol-phase reactions (Carlton et al., 2010).

The following was added to the manuscript at line 101, "In the version of CMAQ used here, new particle formation is based on classical, binary homogeneous nucleation (Kulmala et al., 1998). Particle growth is described by Binkowski and Roselle (2003), and secondary organic aerosol schemes in Carlton et al. (2010)."

3. The results presented in this paper are based on CMAQ simulations with $12 \times 12$km latitude and longitude horizontal resolution, and hourly temporal resolution. As the authors themselves acknowledged: "The 12 km $\times$12 km spatial resolution and 1-hour temporal resolution of CMAQ is significantly larger and longer than the spatial and temporal resolution of the aircraft data, resulting in an inherent contrast in resolution between model and measurement that may play a role in the variance in Sa,mod /Sa,meas". Any reasons why you did not use a nested domain with higher resolution? Also, why not use a higher temporal resolution? 1-hour temporal resolution is too coarse for this kind of simulation (12 km x 12 km).

The goal of this study was to assess how well air quality models that are used for regulatory purposes represent aerosol surface area concentrations. While a finer resolution model may lead to better model-measurement agreement, they are not used as routinely in atmospheric chemistry or as a regulatory model. At the time of submission, the authors did not have access to a nested grid with gridded emissions data to utilize a smaller latitude and longitude resolution than 12km $\times$ 12km, nor was it trivial to reduce the temporal resolution. The same resolution has been used in other work comparing CMAQ to observational data at a finer resolution. For example, Yu et al. (2012) utilizes the same grid scale with hourly resolution in CMAQ compared to observational data from the AIRNow network and aircraft data, Nolte et al. (2015) utilizes the same grid scale and temporal scale compared to four different data sets, and Battye et al. (2016) utilizes the same grid scale compared to a later DISCOVER-AQ campaign. Further, Qin et al. (2018) used a 36km $\times$ 36km grid scale in CMAQ compared to the 2013 SOAS campaign. There is also evidence that the horizontal resolution may not have a large effect on PM comparisons via Wang et al. (2021).

4. Figure 1. Is the figure for an average of values at all altitudes? It will be helpful to separate those in the boundary layer from those above the boundary layer. I expect a large difference in particle size distributions at different altitudes. Also maybe you can use the same y-axis scale for observed and model values for easier comparison?

Figure 1 in the preprint is an average of values at all altitudes. In the revised version of Figure 1, we have included size distributions for below 1km and above 3km for comparison. The aerosol surface area distribution as calculated for altitudes below 1km is similar to that of the full day average originally used, though CMAQ underestimates overall surface area by a larger amount. Above 3km in altitude, the modeled and measured surface area distributions match in shape more closely, although CMAQ overestimates the contribution of larger particles and underestimates those at mid-diameter. The revised figure (shown below) has been updated in the manuscript.

[Figure]

5. The manuscript does not have a summary session. I would suggest that the authors add one.

A summary section has been added to the manuscript (line 525), which reads: "This study examined the ability of the CMAQ model to accurately predict aerosol surface area as it directly affects heterogeneous chemistry within the model. The CMAQ data was compared to dry measured aerosol surface area data from the 2011 DISCOVER-AQ campaign utilizing a UHSAS. A discrepancy between modeled and measured dry aerosol surface area, $S_{a,mod.}$ and $S_{a,meas}$ respectively, are modestly correlated (r2 = 0.52) and on average agree to within a factor of two ($S_{a,mod.}/S_{a,meas}$ = 0.44) over the course of the 13 research flights. However, there was a strong correlation between measured and modeled number concentration ($N_{mod}/N_{meas}$ = 0.87, $r^2$ = 0.63). When looking into possible sources of the discrepancy, there was not a strong dependence on photochemical age or secondary biogenic aerosol concentration. The strong agreement in aerosol number concentration may indicate that the modeled size distribution contributes to the observed discrepancy, though the exact source of discrepancy is outside of the scope of this study.

The discrepancy in aerosol surface area was also compared to that of the reactive uptake coefficient of $N_2O_5$ during the 2015 WINTER campaign due to the fact that the uptake coefficient also directly impacts heterogeneous reaction rates. The uncertainty in the modeled heterogeneous chemistry remains primarily driven by that of the uptake coefficient, as the uncertainty in those values is larger than that seen by $S_a$. Model improvements to aerosol surface area concentrations along with improvements to the parameterization of reactive uptake coefficients will greatly impact the accuracy of heterogeneous chemistry within regional models."

6. L79-81: Are the median diameters for the three modes fixed in the model? How the size range for each mode is decided? How was the value of geometric standard deviation determined?

The median diameters, standard deviations, and size ranges for primary emissions are set within the CMAQ model based on Binkowski and Roselle (2003) from box model inputs. Size ranges are fixed, and modes must always be distinct. Mean diameters and standard deviations are not fixed during particle growth and nucleation. When particle growth and nucleation occur over a timestep, the average mode diameter grows, and some number, surface area, and mass are transferred to the next larger size (e.g. Aitken to accumulation) and averaged with the new size bin. For size distributions and after particle growth/nucleation, these values do not change. The size ranges for each mode are based on Whitby (1978). The geometric standard deviation is also based on Whitby 1978, but have been updated to those from Elleman and Covert (2010) based on many observational data sets.

The following has been added to the manuscript at line 83, "Particle nucleation and growth result in changes to the mode diameter and can result in the transfer of particle number, surface area, and mass to the next larger size (e.g. Aitken to accumulation). Outside of particle growth and nucleation, the approximate median diameters are unchanged. The size ranges for each mode are based on Whitby 1978, and geometric standard deviation is also based on Whitby 1978 but has been updated to those from Elleman and Covert 2010."

7. L100: ">99%": What is the exact value?

Binkowski and Roselle (2003) state that "the major part of $PM_{2.5}$ particulate mass emissions are in the accumulation mode with a small fraction in the Aitken mode; (i.e. 99.9% of $PM_{2.5}$ is assumed to be in the accumulation mode and the remaining fraction, 0.1%, is assigned to the Aitken mode)."

8. L104: How (is the) condensation of water is treated?

As discussed in Binkowski and Roselle (2003), particles acquire liquid water in their evolution, thus increasing the surface area. Particle water concentration is determined using a thermodynamic model (in this case ISORROPIA). ISORROPIA is used both in the "reverse mode" for inorganic gas interactions with coarse particles and in the "forward mode" for inorganic gas interactions with fine particles with AERO6 (USEPA, 2017). Given uncertainty in the contribution of soluble organics to aerosol liquid water content and the lack of "ambient RH" surface area measurements, we focus exclusively on the dry aerosol surface area.

9. Table 1: Q1, Q3 values from Jaegle et al. 2018 appear to be incorrect.

Thank you for catching this. The data was converted from $cm^2/cm^3*10^{-6}$ to $um^2/cm^3$ for the medians, but not the Q1 and Q3. This is updated now.

10. Why did two simulations (2011 & 2015) use two different versions of the CMAQ model and chemistry/aerosol schemes (CB5 vs CB6, AER06 vs AER07)?

Our choice of chemical and aerosol mechanisms has been informed by the speciation of emissions input, which has varied with the development of CMAQ. The 2011 emissions data were speciated for the CB5 mechanism, which is not available in CMAQ v5.3.2. The 2015 emissions were speciated for CB6, which is not available in CMAQ v5.2.1. We expect some differences, as noted in Luecken et al. (2019), "CMAQ-CB6 slightly improves prediction of ozone over much of the dynamic range, while providing updates that are more consistent with current scientific understanding". There are a few other updates from CB5 to CB6, as described in the linked manuscript. On the aerosol side, there is not a distinguishable difference between AERO6 and AERO7 in the source code, and thus aerosol chemistry will not be impacted by the use of AERO7 versus AERO6.

11. L222: Please explain why the measured values are dry Sa.

The *in situ* measurements were made inside the cabin after the ambient air was sampled through an isokinetic inlet. The aerosol temperature equilibrated quickly to the cabin temperature, which is very warm. As a result, the RH of the measurements is likely to be below 40-50% RH. Since the aerosol water associated with particles of this RH is likely to be small (although not negligible) we compare these measurements to the dry model run.

The following sentences have been added to the manuscript (line 237): "Ambient air was sampled through an isokinetic inlet, allowing the aerosol temperature to quickly equilibrate to that of the cabin of the P3-B aircraft. Along with additional ram pressure heating in the inlet during flow deceleration, particles reach an RH of below 40-50%, meaning the aerosol is considered to be dry. The term dry here distinguishes that the particle hydration state is greatly reduced compared to that of the unperturbed ambient air. It is important to note that aerosol water is not accounted for in the following model-measurement comparison, though there would be some aerosol water present at the higher end of the 40-50% RH threshold, which may impact the comparison of the measured aerosol to dry aerosol in the model."

12. L145: relative to?

The large temporal and spatial variability in aerosol surface area in Simon et al. 2010 in the Houston Ship Channel is evident from the larger interquartile range when compared to that in the Gulf of Mexico in the same study (Table 1).

The phrase "relative to the smaller interquartile range seen in the Gulf of Mexico" was added to the manuscript at line 154.

13. L283: Please give some details on how this was measured. Completely dry or at some fixed RH?

Please see our response to comment 11.

14. L305-309: What are the possible reasons? Can the difference be reduced by increasing resolution and/or modifying assumed mode sizes?

As noted in Table 1, the data from Simon et al. and Jaeglé et al. utilized different models. Simon et al. utilized the CAMx model and Jaeglé et al. utilized the GEOS-Chem model. There are a variety of differences between these models and CMAQ, including that CMAQ is a regulatory model on a regional scale while the others are not. The measurements taken by Simon et al. and Jaeglé et al. are also very different to those utilized here. We compare to aircraft data over the Baltimore and Washington D.C. area, while Simon et al. looked at the Gulf of Mexico and the Houston Ship Channel and Jaeglé et al. looked at the northeast US. We compare to these studies because they are two of the only studies that mention aerosol surface area comparisons between models and measurement directly, but we understand that they are not direct comparisons to our data or model utilized in this study.

15. L310: This is inconsistent with what is stated in the figure caption. Please add more labels to the x-axis to make it clearer.

More detail has been added to the figure caption to clarify and the x-axis has been updated for figure 5 a) and b) (below). The figure caption now reads "Figure 5: Histograms of a) measured aerosol surface area concentration (DISCOVER-AQ) at all altitudes, b) modeled aerosol surface area concentration (CMAQ) at all altitudes, c) measured aerosol surface area in the 0 to 1km altitude bin, and d) modeled aerosol surface area in the 0 to 1km altitude bin over the entirety of the DISCOVER-AQ campaign."

[Figure]

16. L388: This is speculation here. Why not use a nested domain with 4 km resolution to look into the effect of resolution?

We did not have access to a nested domain with 4km resolution for the emissions data for this study to be able to look at this directly. We have added a sentence at line 413 to acknowledge that this is a speculation: "However, without utilizing a smaller model resolution to directly test for impacts of the grid size, resolution issues cannot be fully ruled out".

17. L446: Does the condensation change the sizes of particles represented in the model?

Condensation changes particle size by increasing the average particle diameter within a particle mode as particles acquire inorganic liquid water. As the average diameter of the Aitken mode approaches that of the accumulation mode, mass is reassigned to the larger mode, though no more than half of the mass from the Aitken mode can be reassigned during one time step (Binkowski and Roselle, 2003). The particle modes retain the same size range, though the average diameter may change due to condensation.

18. L481: Please explain how the reactive uptake coefficient was measured.

The reactive uptake coefficient was not measured directly. Instead, the uptake coefficient was determined by the method developed by McDuffie where she uses an observationally constrained box model to solve for the $N_2O_5$ uptake coefficient along the flight path of the WINTER campaign.

This information has been clarified in the manuscript in line 507, where the sentence now reads "In Figure 14a, we show the histogram and PDF of the ratio of $\gamma(N_2O_5)_{mod}$ calculated in CMAQ using the Bertram and Thornton parameterization for the WINTER campaign, compared with $\gamma(N_2O_5)_{meas}$, which was determined in McDuffie et al. from an observationally constrained analysis of flight data via a box model solved along the flight path to determine the uptake coefficient (McDuffie et al., 2018)".

19. L485: Why? Winter and summer particle size distribution (and thus Sa) could be quite different.

This is a good point. We have not delved deep into the surface area data for the WINTER campaign, so we are unsure if the surface area is in fact consistent. This statement has been rewritten to clarify and now reads (line 512) "Interestingly, since $k_{het}(N_2O_5)$ is proportional to the product of $S_a$ and $g(N_2O_5)$, the underestimate in model $S_a$ is compensated by an overestimate in $g(N_2O_5)$ in the mean state if the underestimate in model $S_a$ is consistent for WINTER, though that is not necessarily the case".

Responses to comments from Reviewer #2:

1. Figure 13: Since the question is how ratio of modeled to measured aerosol surface area changes with aging, it might be better to plot it with respect to total modeled SOA particle mass rather than just modeled isoprene oxidation products. Can the authors generate a similar plot comparing to modeled SOA?

Similar to the species-specific figure, there is no clear trend with total SOA (as shown below). This figure has been added to figure 13 as part b) of the figure with the original figure 13 as part a) (below).

[Figure]

The statement, "There is also no trend with the total SOA concentration parameterized in CMAQ, shown in figure 13b." has also been included at line 492.

2. Some of the model-measurement differences in aerosol surface area might relate to processes not considered in the model. For e.g.:

(a) How does phase state and viscosity of organic aerosols affect heterogeneous loss of organic aerosols in the model? Heterogeneous loss also affects aerosol surface area and vice versa.

The base version of CMAQ version 5.2.1 does not include phase separation and does not account for changes in viscosity which are known to impact aerosol water and SOA processes. Since we are focused on the dry aerosol surface area these effects could impact total SOA.

Aerosol phase state in CMAQ version 5.2 and beyond is based on ambient relative humidity, according to Pye et al. (2017), where the separation relative humidity (SRH) is the humidity above where a single combined phase state can exist, and "when the ambient relative humidity was below the SRH, the model separated the particle into a water-rich phase (containing aqueous SOA) and an organic-rich phase (containing traditional SOA and POA)". Including organic and inorganic water interactions in absorptive partitioning calculations for organic aerosol generally decreased the bias between CMAQ and measurement data, while increasing the mean error. It is also noted that dryers utilized in aerosol chemistry instruments and different RH systems may play a part in differences between modeled and measured SOA based on phase state calculations, and the same may be said for the data presented in this manuscript, as phase state is important for "wet" aerosol interactions, but cannot be applied directly to dry aerosol, as "Such drying can cause changes in the aerosol phase state and could potentially lead to changes in the partitioning of soluble organic compounds". For example, it has been reported that there can be loss of WSOC after drying, therefore impacting comparisons, though this is impacted by residence time in the specific instrument.

A new algorithm to determine SOA phase separation based on the glass transition temperature, oxygen to carbon ratio, and organic mass to sulfate ratio was applied to CMAQ version 5.2.1, the version run for this manuscript, and evaluated by Schmedding et al. (2020). Adding phase separation to CMAQ version 5.2.1 generally yielded reduction in the uptake of IEPOX to aerosol, thus decreasing the concentration of resultant IEPOX-derived SOA and impacting overall biogenic SOA. It was also predicted that most of the SOA in the middle and upper troposphere was phase separated with more organic concentration in the semi-solid or glassy state as altitude increases, in agreement with field studies. However, this phase-state prediction was not implemented for CMAQ version 5.2.1 as run in this manuscript, so it is possible that the SOA prediction and heterogeneous loss of organic aerosol here is not as accurate as it could be.

The following was added at line 362, "It is also important to acknowledge that some of the model-measurement disagreement could be due to processes not considered in the model such as phase separation, viscosity changes of aerosols, and direct modeling of clouds impacting cloud processing of aerosols, though the impacts of these processes are not investigated further in this work."

(b) How does model representation of aqueous and cloud chemistry affect model-measurement comparisons of aerosol surface area?

Binkowski and Roselle (2003) briefly discusses cloud processing of aerosols in CMAQ. Cloud processes are modeled with simplified effects of clouds rather than modeling the clouds directly. There are a few main assumptions regarding aerosol cloud processing in CMAQ, including:

1) Aitken mode particles are scavenged by cloud droplets, where mass, number, and surface area respond to in-cloud scavenging.
2) Accumulation mode particles form cloud condensation nuclei, which are distributed within the cloud water. Mass, surface area and number can be lost by precipitation, and mass, but not number, is increased by in-cloud scavenging of the Aitken mode.
3) Sulfate mass produced by aqueous production is added to the accumulation mode, but number and geometric standard deviation are unchanged for cumulous clouds.
4) Since the geometric standard deviation does not change for the accumulation mode, surface area is reconstructed with that same geometric standard deviation based on the new mass and number of particles.

5) Aerosol is mixed vertically just as any other species, where wet removal of aerosol is proportional to that of sulfate removal based on the internal mixing of aerosols. Deposition of wet aerosol is removed at the same rate as sulfate particles.

Surface area loss from scavenging of Aitken mode particles and precipitation of accumulation mode particles impacts the wet surface area in the model. Since CMAQ cannot model the clouds directly, and instead interactions within clouds are modeled, it is hard to know whether these processes reflect the actual clouds seen during the campaign used for measurements. Overall, cloud processing focuses more on wet surface area than dry surface area. Since dry surface area is calculated based on the change between wet and dry aerosol volume, and volume does not seem to be considered in cloud processing in the model from this information, we can assume that the impact from cloud processing in our analysis is minimal. However, modeling of cloud processes rather than clouds directly may have an impact on the surface area comparison's accuracy overall.

An in-cloud SOA formation parameterization was added to the CMAQ model at version 4.6. This addition greatly improved the normalized mean bias between CMAQ organic carbon concentrations and those from ground-based and aircraft-based measurements (Carlton et al., 2008). The largest impact was seen on a day when the model was compared to flight data of water-soluble organic carbon specifically designed to investigate clouds (NMB reduced from -65% to -15%), meaning that the addition of the in-cloud SOA formation parameterization included key enhancements to the model for cloud processing.

3. The model-measurement comparisons in the paper are focused on an aircraft campaign that samples up to 5 km altitude (e.g. Fig 8). It would be interesting to compare at higher altitudes e.g. in the upper troposphere as well. Are there measurements from other aircraft campaigns that could be used as well?

We agree that it would be very interesting to look at the model-measurement comparisons at higher altitudes and in more remote regions as aerosol surface area impacts the frequency of NPF. In this paper we focused primarily on the role of aerosol surface area as it connects to ground level air pollution. As such, we sought out a near-surface campaign.

4. How does relative humidity affect model-measurement comparison of aerosol surface area?

As shown below, there is a very slight trend in the surface area ratio with RH in the model for our dry surface area comparison (figure below).

[Figure]

However, it is important to note that we looked at dry $S_a$ measurements and dry modeled $S_a$ in this study. In future analyses, it would be interesting to investigate how the ambient aerosol surface area concentration ratio (model/measured) scales with RH.

5. A general comment: A more useful approach might be to perturb aerosol formation and loss processes in the model e.g. turning heterogeneous chemistry on/off, aqueous chemistry on/off, etc. and determine which of these affect model-measurement comparisons of aerosol surface area the most, especially as a function of aging.

This is a great comment, and we agree that being able to change settings in the model would allow for the ability to find a more detailed/direct reason for what drives the discrepancy between model and measurement. However, doing this would require re-running the model and that is something outside of the scope of this paper. We would love to be able to do so in the future with this data set or a more recent data set to see how each parameter impacts the model-measurement comparison.

6. The reactive uptake analyses only focused on N2O5. Could the authors look at other reactive uptake coefficients as well e.g. related to IEPOX-SOA?

The only measurements we have access to right now are those of $N_2O_5$, which is why that was the focus of the uptake analyses. Other uptake coefficients should be looked at in the future.

References:

Battye, W. H., Bray, C. D., Aneja, V. P., Tong, D., Lee, P. and Tang, Y.: Evaluating ammonia (NH3) predictions in the NOAA National Air Quality Forecast Capability (NAQFC) using in situ aircraft, ground-level, and satellite measurements from the DISCOVER-AQ Colorado campaign, Atmos. Environ., 140, 342–351, doi:10.1016/j.atmosenv.2016.06.021, 2016.

Binkowski, F. S. and Roselle, S. J.: Models-3 Community Multiscale Air Quality (CMAQ) model aerosol component 1. Model description, J. Geophys. Res. Atmos., 108(6), doi:10.1029/2001jd001409, 2003.

Carlton, A. G., Turpin, B. J., Altieri, K. E., Seitzinger, S. P., Mathur, R., Roselle, S. J. and Weber, R. J.: CMAQ model performance enhanced when in-cloud secondary organic aerosol is included: Comparisons of organic carbon predictions with measurements, Environ. Sci. Technol., 42(23), 8798–8802, doi:10.1021/es801192n, 2008.

Carlton, A. G., Bhave, P. V., Napelenok, S. L., Edney, E. O., Sarwar, G., Pinder, R. W., Pouliot, G. A. and Houyoux, M.: Model representation of secondary organic aerosol in CMAQv4.7, Environ. Sci. Technol., 44(22), 8553–8560, doi:10.1021/es100636q, 2010.

Elleman, R. A. and Covert, D. S.: Aerosol size distribution modeling with the Community Multiscale Air Quality modeling system in the Pacific Northwest: 3. Size distribution of particles emitted into a mesoscale model, J. Geophys. Res. Atmos., 115(3), 1–14, doi:10.1029/2009JD012401, 2010.

Kulmala, M., Laaksonen, A. and Pirjola, L.: Parameterizations for sulfuric acid/water nucleation rates, J. Geophys. Res. Atmos., 103(D7), 8301–8307, doi:10.1029/97JD03718, 1998.

Luecken, D. J., Yarwood, G. and Hutzell, W. T.: Multipollutant modeling of ozone, reactive nitrogen and HAPs across the continental US with CMAQ-CB6, Atmos. Environ., 201(July 2018), 62–72, doi:10.1016/j.atmosenv.2018.11.060, 2019.

McDuffie, E. E., Fibiger, D. L., Dubé, W. P., Lopez-Hilfiker, F., Lee, B. H., Thornton, J. A., Shah, V., Jaeglé, L., Guo, H., Weber, R. J., Michael Reeves, J., Weinheimer, A. J., Schroder, J. C., Campuzano-Jost, P., Jimenez, J. L., Dibb, J. E., Veres, P., Ebben, C., Sparks, T. L., Wooldridge, P. J., Cohen, R. C., Hornbrook, R. S., Apel, E. C., Campos, T., Hall, S. R., Ullmann, K. and Brown, S. S.: Heterogeneous N2O5 Uptake During Winter: Aircraft Measurements During the 2015 WINTER Campaign and Critical Evaluation of Current Parameterizations, J. Geophys. Res. Atmos., 123(8), 4345–4372, doi:10.1002/2018JD028336, 2018.

Nolte, C. G., Appel, K. W., Kelly, J. T., Bhave, P. V., Fahey, K. M., Collett, J. L., Zhang, L. and Young, J. O.: Evaluation of the Community Multiscale Air Quality (CMAQ) model v5.0 against size-resolved measurements of inorganic particle composition across sites in North America, Geosci. Model Dev., 8(9), 2877–2892, doi:10.5194/gmd-8-2877-2015, 2015.

Pye, H. O. T., Murphy, B. N., Xu, L., Ng, N. L., Carlton, A. G., Guo, H., Weber, R., Vasilakos, P., Wyat Appel, K., Hapsari Budisulistiorini, S., Surratt, J. D., Nenes, A., Hu, W., Jimenez, J. L., Isaacman-Vanwertz, G., Misztal, P. K. and Goldstein, A. H.: On the implications of aerosol liquid water and phase separation for organic aerosol mass, Atmos. Chem. Phys., 17(1), 343–369, doi:10.5194/acp-17-343-2017, 2017.

Qin, M., Hu, Y., Wang, X., Vasilakos, P., Boyd, C. M., Xu, L., Song, Y., Ng, N. L., Nenes, A. and Russell, A. G.: Modeling biogenic secondary organic aerosol (BSOA) formation from monoterpene reactions with NO3: A case study of the SOAS campaign using CMAQ, Atmos. Environ., 184(December 2017), 146–155, doi:10.1016/j.atmosenv.2018.03.042, 2018.

Schmedding, R., Rasool, Q. Z., Zhang, Y., Pye, H. O. T., Zhang, H., Chen, Y., Surratt, J. D., Lopez-Hilfiker, F. D., Thornton, J. A., Goldstein, A. H. and Vizuete, W.: Predicting secondary organic aerosol phase state and viscosity and its effect on multiphase chemistry in a regional-scale air quality model, Atmos. Chem. Phys., 20(13), 8201–8225, doi:10.5194/acp-20-8201-2020, 2020.

USEPA: CMAQv5.2 Operational Guidance Document, , 1–224 [online] Available from: https://github.com/USEPA/CMAQ/blob/5.2/DOCS/User_Manual/PDF/CMAQ_OGD_Full.06302017.pdf %0Apapers3://publication/uuid/2E2E71FA-F10A-4350-A526-34E61987E2DC, 2017.

Wang, X., Li, L., Gong, K., Mao, J., Hu, J., Li, J., Liu, Z., Liao, H., Qiu, W., Yu, Y., Dong, H., Guo, S., Hu, M., Zeng, L. and Zhang, Y.: Modelling air quality during the EXPLORE-YRD campaign – Part I. Model performance evaluation and impacts of meteorological inputs and grid resolutions, Atmos. Environ., 246(December 2020), 118131, doi:10.1016/j.atmosenv.2020.118131, 2021.

Whitby, K. T.: The physical characteristics of sulfur aerosols, Atmos. Environ., 12, 135–159, doi:10.1016/j.atmosenv.2007.10.057, 1978.

Yu, S., Mathur, R., Pleim, J., Pouliot, G., Wong, D., Eder, B., Schere, K., Gilliam, R. and Rao, S. T.: Comparative evaluation of the impact of WRF-NMM and WRF-ARW meteorology on CMAQ simulations for O3 and related species during the 2006 TexAQS/GoMACCS campaign, Atmos. Pollut. Res., 3(2), 149–162, doi:10.5094/APR.2012.015, 2012.

---

## Author Response (AR2)

For the two parts below, please address the reviewers' comments in a more informative way.

1. The revised text for the response to Review #1 - comment #1 is still unclear. Do you mean all primary emissions use the same size distribution in the model? If so, please provide the size distribution in supporting information as the reviewer requests.

The default parameters for the size distribution of the emitted particles into the CMAQ model are given below. These values originate from paragraph 14 of Binkowski and Roselle 2003 but were updated in 2014 by Kathleen Fahey to reflect those in Elleman and Covert 2010 table 5. These values are located within the CMAQ source code in the "AERO_DATA" module (https://github.com/USEPA/CMAQ/blob/5.2.1/CCTM/src/aero/aero6/AERO_DATA.F).

| Parameter | Aiken | Accumulation | Coarse |
|---|---|---|---|
| dgvem (nm) | 60 | 280 | 6000 |
| def_diam (nm) | 15.0 | 80.0 | 600.0 |
| min_diam_g (nm) | 1.0 | 30.0 | 120.0 |
| max_diam_g (nm) | 80.0 | 500.0 | 100.0 |
| sgem | 1.7 | 1.7 | 2.2 |
| def_sigma_g | 1.70 | 2.0 | 2.2 |
| min_sigma_g | 1.05 | 1.05 | 1.05 |
| max_sigma_g | 2.5001 | 2.5001 | 2.5001 |

Where:
- dgvem = geometric mean diameter by volume,
- def_diam = default background mean diameter for each mode,
- min_diam_g = minimum geometric mean diameter for each mode,
- max_diam_g = maximum geometric mean diameter for each mode,
- sgem = geometric standard deviation of emitted particles in each mode,
- def_sigma_g = default background geometric standard deviation for each mode,
- min_sigma_g = minimum geometric standard deviation for each mode, &
- max_sigma_g = maximum geometric standard deviation for each mode.

The revised text at line 99 has been revised further to read, "Primary aerosol emission rates are provided by the 2011 National Emissions Inventory, which characterizes emissions based on source type and location. Within CMAQ, all primary aerosol emissions, independent of source type, are parameterized with modal size distributions per Elleman and Covert 2010 (see SI)." for further clarification.

The table and explanation are also included in a separate document as supplementary information for the reader should they be specifically interested in the details of the size distribution for primary emissions, as the reviewer requested.

2. The reviewer #1 also asked about how the model scheme for nucleation and growth may affect the modeled size distribution in comment #2, which hasn't been addressed in the response. Please add descriptions about it. For example, how sensitive the modeled size distribution depends on

the parameters used. If other studies have addressed this, the authors may cite as references and summarize briefly their results here to assist the discussion.

As noted previously, the default new particle formation and nucleation parameterization in CMAQ is based on a classical binary sulfuric acid-water homogeneous nucleation from Kulmala et al. (1998). To elaborate on the impact of the model scheme for nucleation and growth on the modeled size distribution, we refer to a comparison of parameterizations for ternary nucleation and nucleation mode processes evaluated by Elleman and Covert (2009). In addition to the binary nucleation parameterization that is utilized within CMAQ, Elleman and Covert investigated a ternary ammonia-sulfuric acid-water parameterization called the Napari parameterization, based on classical nucleation theory with nucleation rates several orders of magnitude higher than the default binary parameterization. The Napari parameterization was utilized in the base case as well as with nucleation mode processing to the Aiken mode to include the number of nucleated particles which survive growth to 10nm and addition to the existing Aiken mode without being lost by coagulation. These parameterizations and the resultant size distributions were compared to observations of size distributions from the coordinated Pacific Northwest 2001 and Pacific 2001 field campaigns. The main impact of changing nucleation parameterization within CMAQ is the large impact of the size distribution below 200nm, where "Adding Napari ternary nucleation increases the prominence of the Aiken mode, but Napari w/Processing better reproduces distinct Aitken and accumulation modes, the peak of the Aitken mode, and the prominence of the Aitken mode relative to the accumulation mode. Above 200nm, changes to nucleation have no effect on the size distribution." However, including ternary nucleation does not solve the issue of reproducing observed size distributions with CMAQ modeling, as nucleation is only one component. The impact of the size distribution from nucleation and growth is evident by this sensitivity study, but the default binary parameterization has not been updated within CMAQ and the addition of nucleation is more complex than simply updating the parameterization.

Though not a direct sensitivity study on the size distribution, Zhang et al. (2010) compared 12 nucleation parameterizations including 7 binary, 3 ternary, and 2 power laws, for their nucleation rates as compared to observations. It is noted that the Napari ternary parameterization (without processing to the Aitken mode) "grossly overpredicts the observed nucleation rates", and the default Kulmala et al. parameterization used in CMAQ has technical mistakes in the formula regarding the kinetic treatment of hydrate formation, reducing nucleation rates. Updating the nucleation parameterization within CMAQ to a more accurate parameterization should be done to improve our prediction of nucleation of particles but is well outside of the scope of this study.

I am also wondering how differ the current understanding about nucleation and growth from the model scheme and whether the difference would affect the modeled results. This is perhaps not the focus of the paper. But it is an important information for readers to understand the results and the interpretation herein. This is somewhat related to the reviewer #2's comment #2(b). But the revised text in line 362 is too brief to strength the discussion.

It is noted by Elleman and Covert 2009 when investigating nucleation parameterizations that "The Kulmala et al. [1998] binary nucleation theory in the standard version of CMAQ v4.4 does

not reflect current knowledge of particle nucleation and processing. More recent versions of CMAQ update portions of its aerosol science but do not change the nucleation code." It is known that the binary sulfuric acid-water theory of nucleation does not fully explain nucleation since the measured sulfuric acid concentrations are not always high enough to produce the observed nucleation rates alone, and there are most likely other processes and species that impact nucleation and need to be further investigated. One of the biggest drawbacks to the default binary parameterization is that there is not a nucleation mode included within CMAQ. The Aitken mode normally is between 10-40nm, while nucleating particles are well below the 10nm threshold, and adding the nucleated particles to the Aitken mode would change the definition of that smallest mode without accounting for loss within particle growth to the traditional Aitken mode. As Elleman and Covert note, "Including nucleation in CMAQ requires both an updated aerosol nucleation theory as well as including nucleation mode dynamics separate from CMAQ's three mode structure.", which has not been implemented at present. Not including nucleation and particle growth of nucleated particles to the Aitken mode impacts the accuracy of the size distributions utilized in CMAQ and therefore the number, surface area, and volume of particles that are predicted. This may explain part of the issue with surface area that was investigated in this paper, though the issue is too complex to pinpoint without further investigation outside of the scope of this study.

The text at line 362 has been updated to clarify and expand on the nucleation and particle growth in CMAQ and now reads, "It is also important to acknowledge that some of the model-measurement disagreement could be due to processes not considered in the model such as phase separation, viscosity changes of aerosols, and direct modeling of clouds impacting cloud processing of aerosols, though the impacts of these processes are not investigated further in this work. The lack of a fourth mode below the Aitken mode for nucleation of particles and growth to the Aitken mode also impacts the accuracy of the size distribution within CMAQ and may explain a portion of the model-measurement disagreement, though it is known that improving the default parameterization does not reduce all errors to the size distribution (Elleman and Covert, 2009b)."

References

Binkowski, F. S. and Roselle, S. J.: Models-3 Community Multiscale Air Quality (CMAQ) model aerosol component 1. Model description, J. Geophys. Res. Atmos., 108(6), doi:10.1029/2001jd001409, 2003.

Elleman, R. A., and Covert, D. S.: Aerosol size distribution modeling with the Community Multiscale Air Quality modeling system in the Pacific Northwest: 2. Parameterizations for ternary nucleation and nucleation mode processes, J. Geophys. Res., 114, D11207, doi:10.1029/2009JD012187, 2009.

Elleman, R. A. and Covert, D. S.: Aerosol size distribution modeling with the Community Multiscale Air Quality modeling system in the Pacific northwest: 3. Size distribution of particles emitted into a mesoscale model, J. Geophys. Res., 115, D03204, doi:10.1029/2009JD012401, 2010.

Kulmala, M., Laaksonen, A. and Pirjola, L.: Parameterizations for sulfuric acid/water nucleation rates, J. Geophys. Res. Atmos., 103(D7), 8301–8307, doi:10.1029/97JD03718, 1998.

Zhang, Y., P. H. McMurry, F. Yu, and M. Z. Jacobson (2010), A comparative study of nucleation parameterizations: 1. Examination and evaluation of the formulations, J. Geophys. Res., 115, D20212, doi:10.1029/2010JD014150.